# Evolution of low-karstified rock-blocks and their influence on reservoir leakage: a modelling perspective

Youjun Jiao 1, 2, 3, 4, Franci Gabrovšek 3, 5, Xusheng Wang 1, Qingchun Yu 1

Correspondence to: Qingchun Yu (yuqch@cugb.edu.cn) and Franci Gabrovšek (franci.gabrovsek@zrc-sazu.si)

**Abstract.** Hydraulic structures such as dams and reservoirs pose significant construction challenges in karst areas due to severe and costly leakage issues. In this study, we apply a numerical model to test the hypothesis that karst aquifers in water divide areas may contain intrinsically low-karstified rock-blocks (LKB), which form due to the specific evolution of unconfined aquifer with recharge distributed to the water table. We develop, test, and apply a model of flow, transport, and dissolution in a 2D fracture network with a fluctuating water table. The model's structure and boundary conditions are based on the conceptualization of the Luojiaao (China) interfluve aquifer. First, we simulate the evolution of an unconfined network, representing the interfluve, up to a stage resembling the present conditions in Luojiaao. We then analyse leakage through the evolved aquifer from a reservoir at different water levels and simulate further aquifer evolution under reservoir conditions. Our results demonstrate the formation of the LKB and highlight its role in mitigating leakage.

#### 1 Introduction

Karst areas occupy approximately 14% of Earth's ice-free land surface. In these regions, the chemical erosion of rock by surface water and groundwater—known as karstification—produces distinctive surface and subsurface features. Karst aquifers are among the most productive freshwater resources, supplying drinking water to around 20% of the world's population—and more than 50% in some countries (Chen et al., 2017; Ford and Williams, 2007). These aquifers are characterized by networks of solution conduits formed within initially fractured and porous rock. The development of these

<sup>&</sup>lt;sup>1</sup>Beijing Key Laboratory of Water Resources and Environmental Engineering, School of Water Resources and Environment, China University of Geosciences (Beijing), Beijing 100083, China

<sup>&</sup>lt;sup>2</sup>Institute of Karst Geology, CAGS/ Key Laboratory of Karst Dynamics, MNR & GZAR/ International Research Centre on Karst under the Auspices of UNESCO, Guilin, Guangxi 541004, China

<sup>&</sup>lt;sup>3</sup>Karst Research Institute ZRC SAZU, Titov trg 2, Postojna 6230, Slovenia

<sup>&</sup>lt;sup>4</sup>Pingguo Guangxi, Karst Ecosystem, National Observation and Research Station, Pingguo, Guangxi 531406, China

<sup>&</sup>lt;sup>5</sup>Faculty of Mathematics and Physics, University of Ljubljana, Jadranska ulica 19, Ljubljana 1000, Slovenia

conduits, a process known as speleogenesis, is complex and spans nearly the entire period during which the rock is exposed to groundwater circulation (Dreybrodt, 1996; Yuan et al., 1993).

Karst aquifers can be conceptualized as hierarchically organized conduit systems embedded within a fractured porous matrix. These conduits serve as high-transmissivity flow pathways, channelling water from points of recharge to karst springs. Karstified rocks are also widespread in young orogenic belts, where the high-relief terrain provides favourable elevation differences that facilitate the construction of hydraulic structures such as dams and reservoirs (Milanović, 2000; Yuan et al., 1993; Shen et al., 1997).




However, the construction hydraulic structures in karst areas pose significant engineering challenges. The position, size, and connectivity of existing networks of solution conduits are extremely difficult to predict or detect. As a result, the extent of initial leakage from a reservoir is also hard to estimate (Milanović, 2000, 2018). Furthermore, the high hydraulic gradients induced by such structures can accelerate the formation of new conductive pathways, potentially leading to a rapid increase in leakage—sometimes reaching intolerable levels within the operational lifespan of the structure (Dreybrodt, 1996; Gabrovšek and Dreybrodt, 2001, 2010; Romanov et al., 2003, 2007).

Basic flow solutions in unconfined porous aquifers with constant recharge as shown in Figure 1a, suggest a relatively stagnant flow zone in water divide area (Rhoades and Sinacori, 1941; T 6th, 1962; Liang et al., 2010). Studies on porous aquifers have revealed a zone of low permeability in divide areas, characterized by slower flow velocities (Kang et al., 2003; Wang et al., 2011; Jiang et al., 2012; Nogues et al., 2013). This also applies to fractured aquifer and can have important implication for karstification, where low flow zone may also result in less karstified zone, as shown in Figure 1b. Such low-karstified rock-blocks (LKB) have been recognised in water divide regions of real karst aquifers and proven to be effective in mitigation of leakage from reservoirs in karst areas (Yuan et al., 1993; Milanović, 2000; Xu and Yan, 2004). The LKB means that there is a relatively low permeable region formed in the middle of karst aquifer during the natural karst evolution. As shown in Figure 1c, a low-karstified zone with an extremely low velocity is pivotal in understanding karst aquifer leakage in water divide areas. After construction of a reservoir, if the groundwater divide persists, aquifer leakage will be absent, significantly minimizing the need for anti-seepage measures.

Figure 1. Simplified cross section of aquifer in water divide region. (a) The flow solution reveals the presence of a stagnant zone near the groundwater divide within the porous aquifer. (b) Karstification increases permeability and progressively lowers the water table over time, leading to the formation of highly karstified rocks and low-karstified rock-blocks (LKB). (c) When reservoirs are constructed, the LKB can effectively obstruct leakage across the aquifer. (Modified from Zhang et al., 2018).


Due to the intricate nature of karst leakage, there have been limited reports on quantitative modelling studies of this phenomenon (Milanović, 2000, 2018; Smerdon et al., 2005). Modelling karst aquifer leakage is a challenging endeavour, often deemed impractical despite its importance for engineering planning. In engineering construction, predictions of

reservoir leakage primarily rely on geological surveys and field experiments. For instance, one study analysed extensive observed data and forecast that leakage would remain minimal (Yuan et al., 2002; Shen et al., 1997).

To model evolution of karst aquifers, it is necessary to simulate flow, dissolution, and transport processes within soluble fractured media. Modelling an unconfined aquifer with a water divide also presents a free surface problem, requiring the determination of the water table position at each model time-step. Some studies have addressed this by treating discrete fractures as equivalent continuum media (Desai and Li, 1983). However, these models cannot simulate fracture dissolution and widening. Several studies have reported on modelling discrete fracture flow with free surfaces. Wang (1993) employed an initial flux method to locate the free surface within the fracture system, while Jing et al. (2001) used a discontinuous deformation analysis (DDA) method for modelling fracture flow. These intuitive methods often encounter numerical instability due to their lack of a solid mathematical foundation.




Several models have been employed to simulate the evolution of unconfined karst aquifers with constant recharge of the water table (Clemens et al., 1999; Gabrovšek and Dreybrodt, 2001, 2010, 2021; Kaufmann, 2003). However, these models often struggle with large-grid aquifers or those containing randomly distributed heterogeneous fractures. In intuitive approaches, testing whether fracture nodes near the water table are wet or dry frequently results in inconsistencies, with nodes oscillating between being below and above the water table across iterations. This suggests that modelling the water surface under rainfall recharge may require a revised approach. The variational inequality method, which introduces the continuous Heaviside function in the finite element method, has proven effective in modelling the free surface in discrete fractures (Zheng et al., 2005; Jiang et al., 2013). But these models do not consider situations with recharge to the water table. Therefore, in the context of random fractured aquifers, a simplified and fundamental approach to modelling the free surface under rainfall conditions is essential to capture complex flow and dissolution dynamics.

As demonstrated by the models of Gabrovšek and Dreybrodt (2001) and Kaufmann (2003), the evolution of unconfined karst aquifers is most pronounced near the water table. In brief, these models do not explicitly account for the details of flow and dissolution in the vadose zone—the region between the surface and the water table—but instead assume gravitational flow that reaches a specific level of chemical saturation with respect to calcite. Nevertheless, within the phreatic zone, water is typically least saturated with respect to calcite near the water table, and thus possesses the highest potential for

dissolution (often referred as aggressivity) in this region. The numerical methods employed in these studies primarily concentrate on the evolution of relatively homogeneous fracture networks, rather than the random fractures found in natural karst aquifers.

In this work, we introduce a numerical model to study the evolution of karst aquifer under distributed (rainfall) recharge. The algorithm is built upon an earlier study by Yu et al. (1999). We propose an intuitive method that uses iterative search for the water table and couples flow, transport and dissolution to simulate evolution of discrete fracture network. We first simulate the natural evolution of the aquifer under rainfall recharge up to selected stage of maturity. Then, the obtained aquifer is subjected to boundary conditions imposed by the construction of the reservoir. We analyse how the LKB has evolved and assess its effectiveness in mitigating leakage from the reservoir.

# 2 Modelling evolution of unconfined discrete fracture network



Models of fracture networks' karstification necessitate the integration of flow, dissolution, and transport processes within the system, while adhering to specified hydraulic and hydrochemical boundary conditions. In our research, we simulate the evolution of a stochastic fracture network under constant recharge and fixed head boundary conditions. The computational procedure is shown in flowchart of Figure 2 and outlined as follows:

- (1) Generate random fracture network and boundary conditions;
- (2) Locate the water table and compute the flow within the network;
- (3) Calculate dissolution and solute transport;
- (4) Update fracture apertures: compute the changes in the width of each fracture within the specified time step;
- (5) Iterate or terminate: proceed to the next time step and repeat steps 2 to 4, or terminate the simulation as required.

**Figure 2.** The flowchart of the whole modelling process. At each time step, the new position of the water table is determined through an iterative process. Subsequently, the coupled flow, dissolution, and transport equations are solved at both the individual fracture scale and the fracture network scale. Based on these results, the fracture apertures are updated accordingly.

The modified fracture network then serves as the basis for calculations in the following time step.

#### 2.1 Flow in fractures

#### 110 **2.1.1 Fracture network generation**

The initial fracture network is created through Monte Carlo stochastic simulation, as described by Yu et al. (1999). Fracture parameters, including centre location, length, aperture, and direction, are randomly generated using Gaussian and lognormal distributions. Subsequently, the algorithm searches for fracture intersections (nodes) and excludes all dead end segments, which do not contribute to flow. In this work the fractures are uniform plan parallel planes, defined by length and aperture.

#### 2.1.2 Flow in one fracture


When a fracture with length  $L_{ij}$  and aperture  $b_{ij}$  connects nodes i and j with hydraulic heads  $H_i$  and  $H_j$ , respectively, the flow rate  $q_{ij}$  within the fracture conforms to the cubic law for laminar flow, whereas for turbulent flow, the Lomize equation (Lomize, 1951) is used to relate the flow rate to the hydraulic gradient  $J_{ij}$ .

$$q_{ij} = \frac{g \, b_{ij}^{\ 3}}{12 \, v} \, J, \quad Re < 2020 \tag{1}$$

$$q_{ij}=4.7 \sqrt[7]{\frac{g^4}{v}b_{ij}^{12}J^4}, \quad Re>2020$$
 (2)

$$J_{ij} = \frac{H_i - H_j}{L_{ii}} \tag{3}$$

where g is the gravitational acceleration, v is the kinematic viscosity of water, and Re is Reynolds number. The transition from the laminar to the turbulent flow regime is here introduced at the point where turbulent frictional losses exceed those associated with laminar flow. The factor in Lomize equation (Eq. 2) is set so that transition occurs at a Reynolds number around 2020. This simplification omits the transitional flow zone, resulting in a smooth increase in flow rates as turbulence sets in (Su et al., 1994; Finenko and Konietzky, 2024). However, in simulations presented in this work, flow always remains laminar.

# 2.2 Solution for water table in random fractures under distributed uniform recharge conditions

Each evolution time step requires the calculation of flow, dissolution, solute transport, and the corresponding changes in fracture aperture. Accurate flow calculation depends on determining the position of the water table at each time step. This is achieved through an iterative process, in which the water table position is updated until specific convergence criteria are met, as described in Section 2.2.2. The procedure is illustrated in Figure 3. At each iteration, the current approximation of the water table defines a set of boundary conditions (Figure 3a): prescribed recharge at water table nodes, seepage face (boundary nodes, where hydraulic head equals the elevation), and constant-head conditions elsewhere. The flow solver (confined flow solution) is then invoked to compute flow rates and hydraulic heads at all nodes. Water table nodes are subsequently searched node by node within each layer (Figure 3b, node-by-node iteration) and layer-by-layer throughout the domain (Figure 3c, layer-by-layer iteration).

**Figure 3.** Conceptual presentation of flow calculation in the fracture network. (a) Boundary conditions for confined flow calculation performed at every iterative step. (b) Node-by-node iteration: testing wetting of a dry node across a layer. (c) Layer-by-layer iteration.

#### 140 2.2.1 The solution of the confined water flow with rainfall recharge

At each node-by-node iteration, an approximate water table classifies boundary nodes as having a specified head, prescribed recharge, or belonging to the seepage face. At each node, the flow balance is required:

$$\sum q_{ij} + \sum q_w = 0 \tag{4}$$

Where the first part is the sum of flow from/to all inner (non-boundary) nodes (see Eqs. 1-2) and the second part is the sum of direct recharge  $q_{\rm w}$  from the vadose zone, applied to the water table nodes.

Assuming that the equations are non-linear, we use the Newton–Raphson iteration to solve the system of equations, with convergence criteria as follows:

$$\Delta H_{\text{max}} \le H_{\text{tol}}, \text{ for } \Delta H_{\text{max}} = \max\left(|\Delta H_{\text{error}}|\right)$$
 (5)

$$Q_{\text{erro}} \le Q_{\text{tol}}, \text{ for } Q_{\text{erro}} = \sum q$$
 (6)

where  $\triangle H_{\text{error}}$  is the water head error vector between the n and n+1 iterations. H is the water head vector.  $\triangle H_{\text{max}}$  is the maximum water head difference between two iterations and  $H_{\text{tol}}$  is the allowable error.  $Q_{\text{erro}}$  is the sum of water flow budget of all inner nodes, and  $Q_{\text{tol}}$  is the allowable error.

The difference in the water heads between two iterations and the water budget is checked to validate the flow results. When  $\triangle H_{\text{max}}$  is less than the allowable error  $H_{\text{tol}}$  and the water budget  $Q_{\text{erro}}$  is less than the allowable error  $Q_{\text{tol}}$ , the iteration converges. The C++ open source linear algebra package Armadillo (Sanderson and Curtin, 2016, 2019) was employed to perform matrix computation.

#### 2.2.2 The node-by-node and layer-by-layer iterations: search for the water table

155

One of the challenges in modelling unconfined aquifers with distributed recharge lies in accurately identifying the water table. In the presence of dissolution and increasing permeability of fractured media, the water table changes in time

also when recharge is constant. We use a robust "intuitive" algorithm for computing the position of a water table as shown in Figure 3. The algorithm is applied for all time-steps of a simulation.

**Step 0:** Initially, the water table is positioned at the lowest constant head boundary in the domain (Figure 3a). The nodes of the water table receive direct recharge from the vadose zone. The heads at all nodes in the phreatic part are calculated by above described confined flow solution.

- Step 1: Dry nodes, directly connected to water table nodes are identified. For each pair, the difference between the hydraulic head of the nearest water table node  $H_i^{WT}$  and the elevation of the dry node  $z_i^D$  is calculated as  $\Delta_{ij} = H_i^{WT} - z_i^D$ . These differences are then sorted into an ordered sequence.
- Step 2: Following the sequence each dry node is changed to a water table node and the heads are recalculated by confined flow solver, assuming new boundary conditions with constant input to the new WT node (Figure 3b). If the head at the node is higher than its elevation, and no other wet node becomes dry, the transition of the node from dry to wet is kept. Otherwise, the node remains dry and the procedure continues with the next node in the sorted sequence.
  - **Step 3:** Repeat Steps 1 and 2 layer by layer until no dry node becomes wet (Figure 3c).

The algorithm employs a two-level iterative strategy: (1) layer-by-layer iterations and (2) node-by-node iterations within each layer. When the confined water head iteration is included, the method effectively operates as a three-level iterative strategy, ensuring robustness and accuracy in resolving wetting and free surface dynamics.

#### 2.3 Dissolution widening of fractures


The dissolution rates of limestone under both laminar and turbulent flow conditions have been the focus of extensive research (Dreybrodt, 1990, 1996; Liu and Dreybrodt, 1997; Eisenlohr et al., 1999; Dreybrodt and Kaufmann, 2007). These 175 rates are governed by a complex interplay of surface reactions, transport mechanisms, and CO<sub>2</sub> conversion processes (Dijk et al., 2002; Detwiler and Rajaram, 2007). In this study, we adopt the rate equations developed by Dreybrodt (1996):

$$F(c) = k_1 \left( 1 - \frac{C}{C_{eq}} \right), \quad (C < C_s)$$
 (7)

$$F(c) = k_1 \left( 1 - \frac{C}{C_{eq}} \right), \quad (C < C_s)$$

$$F(c) = k_n \left( 1 - \frac{C}{C_{eq}} \right)^n, \quad (C > 0.9C_s, n > 1)$$

$$(8)$$

Where  $k_1$  and  $k_n$  are rate constants (in mol cm<sup>-2</sup> s<sup>-1</sup>), C represents the concentration of Ca<sup>2+</sup> ions (in mol L<sup>-1</sup>), and  $C_{eq}$  denotes the equilibrium concentration in the H<sub>2</sub>O-CO<sub>2</sub>-CaCO<sub>3</sub> system. The  $C_{eq}$  can be calculated from the concentration of calcium and dissolved  $CO_2$  of the solution at the water table. This equilibrium state results from complex flow and dissolution processes occurring in the vadose zone, which are beyond the scope of this study. In this work we take uniform equilibrium concentration (2 mmol/L) and uniform saturation ratio ( $C_{in}$  = 0.92 $C_{eq}$ ) at water table nodes. The reaction follows a linear rate law up to the switch concentration  $C_s$ , transitioning to a nonlinear rate law between  $C_s$  and  $C_{eq}$ . Here we assume that rates are entirely controlled by surface reaction, and ignore the concentration gradient perpendicular to flow. This approximation is valid for situations where the solutions are close to equilibrium (Gabrovšek, 2000), which is mostly the case in the presented scenarios. The reaction order n and the switch concentration  $C_s$  are experimentally determined and influenced by the impurity content in the limestone (Eisenlohr et al., 1999). Experimental studies have reported reaction orders ranging from 3 to 11. For this study, we use n = 4,  $C_s = 0.9C_{eq}$ ,  $k_1 = 4 \times 10^{-11}$  mol·cm<sup>-2</sup>·s<sup>-1</sup>, and  $k_4 = 4 \times 10^{-8}$  mol·cm<sup>-2</sup>·s<sup>-1</sup>(Gabrovšek and Dreybrodt, 2000; Dreybrodt et al., 2005). Studies on the dissolution kinetics of calcium carbonate minerals under turbulent flow (Buhmann and Dreybrodt, 1985a, 1985b; Liu and Dreybrodt, 1997) indicate that the rate constant in turbulent conditions is an order of magnitude greater than that under laminar flow.






The change in concentration  $\Delta C$  along an individual fracture can be determined by applying the principle of mass conservation within a water parcel passing along the fracture. This can be easily analytically calculated for parallel plan (the flow perimeter P(x) is constant) fracture, as shown in the Supplements. Change of concentration along the fracture is converted to mass removed from the fracture walls and the change of aperture during time  $\Delta t$ :

$$\Delta b = \frac{\Delta C \ q \ M_{\text{CaCO}_3}}{\rho \ L} \Delta t \tag{9}$$

Where L is the fracture length,  $\rho$  is the density of calcite (2.5 g/cm<sup>3</sup>),  $M_{CaCO_3}$  is the molar mass of calcite (100 g/mol), and q is the flow rate.

To calculate the change in concentration along a fracture, the calcium concentration at the input node must be known. This input concentration is determined as the flow-weighted average of all inflows to that node. To ensure that all required concentrations are available when needed, the computation proceeds in a cascading sequence, starting from nodes with the highest hydraulic heads and moving downstream (see the Supplements).

Uniform widening with average widening rate within a fracture is assumed. When the fracture network's aperture is updated due to dissolution widening, the current evolution time step is completed. The process then advances to the next time step, during which the water head and solute concentration are recalculated.

#### 205 **2.4 Model verification**



To verify the numerical model, we first compare the results for the uniform network with MODFLOW (Harbaugh et al., 2000) and the analytical solution derived with the Dupuit assumption. The water tables simulated with our method are nearly the same as the MODFLOW simulation. Then we test the solution for heterogeneous network. The seepage faces above the constant head boundaries on the both sides of the domain are successfully simulated. The analysis demonstrates the effectiveness of our algorithm. The details of the verification are shown in the Supplements.

#### 3 Modelling karstification in an interfluve aquifer with rainfall recharge

As an example of an aquifer, where the concept of LKB formation can be applied is Luojiaao interfluve aquifer, located between the two reaches of the Qingjiang river meander. Hubei Province, China (Figure 4a and b). On the western side of the interfluve, the Geheyan Reservoir was constructed, which is one of three karst reservoirs along the river (Xu and Yan, 2004). The water level in the reservoir is above the original water divide, which was determined by boreholes (Figure 4c). However, the leakage and its trend could not be determined with field methods.

Although we lack knowledge on the possible previous karstification stages at this particular site, we start with a naive assumption, that the water divide in the interfluve is a result of karstification under rainfall recharge, and that it can be used as an example of a site, where the LKB may play a crucial role in reservoir leakage.

**Figure 4.** Location and setting of the Geheyan Reservoir, situated on the upstream side of a bend in the Qingjiang River, Changyang County, Hubei Province, China. (a) The location of Geheyan Reservoir. (b) Simplified map of the reservoir area and Luojiaao interfluve aquifer (marked by a dashed rectangle in figure a). (c) Cross-section through the Luojiaao interfluve aquifer along the line AB in figure b. The dashed line shows position of the water table before the reservoir construction.

# 3.1 Hydrogeological basis and conceptual model



The conceptual model, which is translated into numerical one, is based on simplification of field mapping results. We assume a limestone aquifer with sub horizontal dip (8–10 °) and set of vertical fractures with a mean dip angle of 85 °. Past studies of the Luojiaao interfluve aquifer (Yuan et al., 2002; Shen et al., 1997; Wan et al., 1999) have revealed caves close to

the river bed on both sites and absence of caves in the water divide region. It is difficult to reconstruct past hydrogeochemical boundary conditions at the water table. Based on the results of soil  $CO_2$  and  $Ca^{2+}$  content in three boreholes and two springs, we assume that the closed system  $C_{eq}$  is 2 mmol/L.

Table 1 Statistical parameters of the random fracture network

| group | parameter      | distribution | mean | standard deviation | min   | max   |
|-------|----------------|--------------|------|--------------------|-------|-------|
| 1     | direction ( °) | Gaussian     | 11   | 1                  | 9     | 13    |
|       | length (m)     | lognormal    | 240  | 5                  | 220   | 260   |
|       | aperture (cm)  | uniform      | 0.01 | /                  | 0.008 | 0.012 |
| 2     | direction ( °) | Gaussian     | 85   | 2                  | 82    | 88    |
|       | length (m)     | lognormal    | 240  | 5                  | 220   | 260   |
|       | aperture (cm)  | uniform      | 0.01 | /                  | 0.008 | 0.012 |

The modelling domain as shown in Figure 5a is a two-dimensional fractured limestone aquifer with a length of 2000 m and a height of 500 m. The two groups of fractures and joints were generated randomly according to the statistical parameters listed in Table 1. We assume the uniform distribution of initial apertures with the expected value of 0.01 cm. There are totally 4073 intersection nodes and 7401 fracture segments. Constant head of 100 m is taken at left and right side of the domain, between elevations -100 to 100 m and seepage boundaries between 100 m and 400 m. Constant infiltration of 200 mm/a is applied uniformly to the water table nodes. We assume that the saturation ratio ( $C_{\rm in}$  /  $C_{\rm eq}$ ) of water entering the 240 phreatic zone is 0.92. The simulation time was set to 100 thousand years. Model convergence was checked at every evolution time step.

## 3.2 Model of karstification under distributed uniform recharge

Figure 5. The initial aquifer (a), and aquifer after 10 ka (thousand years) (b) and 20 ka (c) of evolution. The widths of the lines present the aperture, colours the head and arrows the flow direction. To facilitate the discussion the aquifer is divided into 3 primary vertical sections and each of them into 5 horizontal subsections, as shown in panels a and d.

The modelling results are shown in Figure 5. We observe the expected drop of water table due to increasing transmissivity of the aquifer. The groundwater divide remained close to the middle of the domain, whilst the elevation of the highest wet node, which represents the divide ridge, descended from 350 m to 230 m at 10 ka and to 185 m at 20 ka.

As expected, there is a fringe of high dissolution rates close to the water table. Below the water table, the fracture widening is relatively slow. As the water table gradually decreases, the fringe of high dissolution rates migrates vertically through the aquifer and increases the hydraulic conductivity.

# 3.3 Changes in fracture aperture and discharge during karstification


**Figure 7.** The box-whisker plot of fracture aperture under natural evolution for 20 thousand years in three primary sections (see Figure 5).

The fracture aperture distributions at three evolution time steps are shown in Figure 6. To facilitate the discussion, we have divided the aquifer into 15 subsections along three vertical primary sections, as shown in Figure 5a. The variation of aperture statistics in three vertical sections is shown in Figure 7. The water table with dissolution fringe mainly descends through the upper section, which experience the evident change in aperture (Hubinger and Birk, 2011).

Figure 8. The relative change in the average aperture  $b/b_0$  and the change in discharge rate q (flow out) in different subsections.

The changes of apertures and discharge rates in several representative subsections are shown in Figure 8. Generally, the change of apertures drops along vertical directions and is higher near the left and right boundary. In the lower section, the change of aperture is minimal (A13).


The discharge rate in each subsection was computed by checking the fractures that cross the boundary and flow into the subsection. The lateral aquifer width was assumed to be 100 m. The A13 subsection has the lowest discharge, which changes the least over time. The discharge in the A8 subsection changed more slowly than that in other parts of the middle primary section. The A3 subsection, in the middle part of the upper primary section, exhibited a much greater

leakage trend than did the lower parts. From the aperture and discharge analysis, we concluded that a low-karstified area exists mainly in the middle and lower primary sections.

Karstification represents a form of nucleation, where flow-induced dissolution and changes in porosity are coupled through feedback mechanisms (Edery et al., 2021; Molins et al., 2014). In unconfined aquifers under constant recharge conditions, dissolution is most intense near the water table. This process creates a highly permeable fringe that effectively channels inflow toward both sides of the water divide. As this fringe migrates downward across the aquifer cross-section, it leaves behind a distinctive porosity imprint. Simultaneously, it inhibits deeper penetration of the inflowing solution, favouring the preferential development of horizontal fractures. Moreover, flow along the water table increases progressively from the water divide toward the discharge points. As a result the water divide zone is less karstified than the regions close to the output. Similar anisotropic, directional changes, including fingers or preferential flows, has also been observed through experimental studies and other numerical simulations (Rege and Fogler, 1989; Shavelzon and Edery, 2022; Singurindy and Berkowitz, 2003).

# 3.4 Changes in conductivity K and formation of the LKB






To further elucidate the formation of an LKB, we calculated vertical and horizontal hydraulic conductivity in each subsection. To do this, we "cut out" a section at certain stage of evolution, and calculated confined flow rate for a given head difference along horizontal and vertical direction, respectively. Thus horizontal and vertical equivalent K values, and the changes and trends in the representative subsection were computed (Figure 9). Notably, the fractures below 50 m in the lower subsections (A11 to A15), remained unevolved. The horizontal K varied between 0.005 and 0.008 m/d in the different subsections, and the vertical K varied between 0.001 and 0.002 m/d.

The middle subsections A7, A8, and A9 experienced low level of karstification, in particular, subsection A8 (less than 0.01 m/d), which combined with the lower section forms an LKB, which could effectively prevent horizontal flow. In the A6 and A10 subsections, the K increased by more than one order of magnitude.

The upper primary section was also tested to determine whether it included a low permeable part. Note that here only phreatic karst process is considered, therefore most fractures in subsection A1 did not experience widening. The water table descended through subsection A3, where the conductivity trends increased in both the horizontal and vertical directions

compared with those in the other subsections. Although the upper primary area does not exhibit a low conductivity, its underlying low permeable rock-blocks have already been recognized as an LKB.

**Figure 9.** Changes and increasing trends of the equivalent conductivity K in the different subsections during 20 ka natural evolution process.

# 4 The influence of LKB on reservoir leakage




To verify the relevance of LKB formation on reservoir leakage, as in the case of Luojiaao interfluve, we now apply reservoir boundary conditions to an evolved aquifer. The borehole data revealed that the natural water divide level is 50 to 80 meters above the river, which roughly corresponds to the situation in our synthetic aquifer after 20 ka of evolution.

# 4.1 Reservoir leakage with various reservoir levels

First, we test the leakage through the aquifer at different reservoir water levels. Therefore, we apply constant head of 200 m, 250 m and 300 m, on the left boundary of the aquifer (presenting reservoir), which has evolved for 20 ka. The flow fields are shown in Figure 10. In case of 200 m, the groundwater divide still exists, although it is shifted towards the left boundary. However, the water still flows from the aquifer to the reservoir. When the water is raised to 250 m, as shown in Figure 10b, the groundwater divide moves almost to the left boundary, so that the flow to the reservoir is only local (upper left), however, deeper the flow is directed from the reservoir to the aquifer. At 300 m, as shown in Figure 10c, the flow direction is almost entirely from the reservoir to the aquifer.

**Figure 10.** Flow fields within the aquifer developed over the 20 thousand years), after the application of reservoir boundary conditions on the left side at: (a) 200 m, (b) 250 m, and (c) 300 m.

Figure 11. Leakage as a function of reservoir level for aquifer after 20 ka, 30 ka and 40 ka of evolution. The lateral width is 100 m.

Figure 11 shows the reservoir leakage at different water levels through an aquifer at different stage of evolution, assuming the lateral width of the aquifer is 100 m. For aquifer at 20 ka, no leakage is observed below 225 m, as the water divide is still present and high enough to drive water from the aquifer to the reservoir. At 30 ka and 40 ka, leakage is present at all levels applied and increases linearly with the reservoir level. The actual water level of the Geheyan reservoir is 200 m, which is below the level, where significant leakage may be expected from the model.


# 4.2 LKB changes before and after reservoir filling



**Figure 12.** Variation of hydraulic conductivity (K) in the karst aquifer before and after reservoir filling: horizontal (a) and vertical (c) distribution of hydraulic conductivity after 20 ka of natural evolution; horizontal (b) and vertical (d) distribution of K after one thousand years of reservoir conditions. The saturation ratio  $(C/C_{eq})$  of reservoir water is 0.8.

If water in the river is aggressive to calcite, evolution of aquifer continues after the reservoir filling. We assume that the saturation ratio  $(C / C_{eq})$  of reservoir water is 0.8. Figure 12 shows the distribution of hydraulic conductivities in an aquifer, which has evolved naturally for 20 ka and then under the influence of reservoir for one thousand years. The subsections A8 and A13 maintained the lowest subsections of conductivity after reservoir filling. From the conductivity

analysis of the middle and lower primary sections, we can conclude that the formed LKB remained a continuous and low permeable wall under one thousand years of reservoir filling conditions.

**Figure 13.** Reservoir leakage through karst aquifers over one thousand years (assuming that the aquifer lateral width is 100 m).

Figure 13 illustrates the increasing trend of reservoir leakage over a 1,000-year period of reservoir operation. Leakage rates were estimated assuming a lateral aquifer width of 100 m. At the 200 m reservoir level, no leakage occurs initially, resulting in no further evolution over time. For the 250 m and 300 m levels, leakage remains below 2 L/s over the entire 1,000-year span.

While the severity of leakage under these idealized conditions may appear acceptable during long-term reservoir operation, the final determination of a permissible water level must consider a broader range of factors. These include geological uncertainties, site-specific topographic features, and social safety requirements, all of which play critical roles in the decision-making process.



The results show moderate and acceptable increase of leakage within the expected life span of the dam (about 100 years). However, we have to be aware that model is idealisation of reality, and that further structural, speleological and hydrological data would be required to give a more reliable site-specific prediction. In practice, when the numerical model has been sufficiently calibrated and verified with enough field data, such as observed fractures distribution and

groundwater levels, the prediction of leakage is more reliable. Then, the magnitude and changing patterns of the leakage provided from the model can be used to optimize the reservoir water level and other engineering measures (Zhou and Li, 1996).



The fracture flow velocity distribution during the natural condition evolution process and under different water levels of the reservoir is shown in Figure 14. Compared to other regions, the presence of areas with low flow velocity in the central part indicates low permeability and weak karstification characteristics. The high-velocity low forms an arch-shaped region along the permeable fringe, which has developed along the descending water table. In the water divide region, the LKB forms, mitigating flow beneath the fringe. Under reservoir conditions, leakage becomes substantial only when the reservoir level rises well above the LKB, causing the water divide to shift closer to the reservoir and initiating flow along the permeable arch.

**Figure 14.** Velocity distribution in the aquifer at different stages and conditions: (a) at the onset of karstification, (b) after 20 ka of natural evolution prior to reservoir impoundment, (c) after 20 ka of natural evolution and then with a reservoir level of 200 m, and (d) after 20 ka of natural evolution and with a reservoir level of 300 m.

#### 5 Discussion



**Figure 15.** Conceptualisation of the low-karstified rock-blocks (LKB) in the water divide region (shaded area between dotted lines b and c). The zone below line c exhibits minimal karstification. Line a indicates the naturally evolving water table prior to reservoir impoundment, with conduits and caves forming on both sides of this water level.

The concept of LKB, also demonstrated by our model, is summarized in the sketch map in Figure 15. The dotted line marked with "a" is the evolving water table under natural conditions. Below line c, the fractures have experienced almost no karstification. Between line a and b, well karstified regions are developed on both sides, with the two sides of the water table descending area. However, in the area between line b and line c, the fractures are widened only to a certain degree. The area between lines b and c is an LKB, which presents a natural low permeable zone, which mitigates potential leakage from the reservoir.

Reservoir leakage is highly sensitive to the water level relative to the position of the LKB or the water divide. Therefore, identifying the position of the water table in the aquifer prior to reservoir impoundment can help infer the location of the LKB and potentially reduce the need for costly engineering measures aimed at leakage prevention.

Karst aquifers evolve in different climatological, hydrological and geological settings. Additionally, in active tectonic areas, the structure and boundary conditions may change during the karstification, which may continuously adapt to the terrain uplift and/or valley entrenchment. Therefore, any modelling related to a specific site must include all information on the environment in which an aquifer has evolved (Class et al., 2021; Hartmann et al., 2013). If the aquifer is uplifted due to

deformation, the uplift rate and the river trenching rate influence the extent of the relatively low permeable rock-blocks (Gabrovšek et al., 2014).

# **6 Conclusions**





The presented model simplifies the system by neglecting many of the complexities observed in natural settings. In reality, karst aquifers often undergo multiple phases of karstification over geological timescales, driven by varying hydrological and geochemical boundary conditions. The initial structure of an aquifer is typically far more intricate than assumed in the model, potentially including pre-existing preferential flow paths, geochemical heterogeneity (e.g., interbedded layers of less soluble rock), and temporal variability in recharge.

These features could be systematically incorporated into the model to investigate their influence on aquifer evolution.

Additionally, the model does not account for the development of the vadose zone, which plays a crucial role in modifying the permeability of layers above the water table and influences the geochemical characteristics of the water at the water table.

Despite its limitations, the model effectively serves as a proof of concept, offering valuable insights for hydraulic engineering in karst regions. Many karst aquifers have developed through natural karstification under conditions of distributed recharge. As demonstrated in this study, such evolution leads to a characteristic permeability structure: zones of enhanced conductivity form along the descending water table, while the low-karstified rock-blocks (LKB) persist beneath the water divides.

We used the Luojiaao interfluve as a representative example to illustrate how the concept of LKB can explain the notably low leakage rates from the reservoir impounded along the boundary. These low karstified rock blocks, being less permeable, act as barriers to flow and play a crucial role in controlling groundwater movement and reservoir interactions. The recognition and modelling of the LKB thus provide a powerful framework for understanding and managing water resources in complex karst systems.

## **Data Availability Statement**

All data used in this research are publicly available. The data and code used for simulating fracture phreatic water table and karst evolution are available at https://github.com/jiaoyj/Fracturetokarst2024.

## **Author contribution**

YJ contributed with writing of the original draft, methodology and modifying the code. FG contributed with conceptualization, review and editing of the manuscript. XW contributed with conceptualization, review and editing of the manuscript. QY contributed with the original code, conceptualization, methodology and review.

**Competing interests.** The authors declare that they have no conflict of interest.

# Acknowledgments

This work was supported by the National Natural Sciences Foundation of China (Grant Nos. 42377061, 41772249, 41702279), Guangxi Key R&D Project (AB25069394), Guizhou Academician Workstation (KXJZ[2024]005), China Geological Survey (DD20251205) and the China Scholarship Council. The authors are very grateful for the help of the Rock Fracture Work Group. The work of Franci Gabrovšek was supported by the research program Karst Research (P6-0119(B)), funded by the Slovenian Research and Innovation Agency (ARIS), and by the project ERC KARST (101071836), funded by the European Research Council.

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
