# Peer review of "Evolution of low-karstified rock-blocks and their influence on reservoir leakage: a modelling perspective"

_EGUsphere, 2025_

## Author Comment (AC1)

**REPLY TO THE REVIEWER'S COMMENTS**

We sincerely appreciate your thorough and thoughtful review. Many of the comments and suggestions required careful consideration and prompted substantial revisions to the manuscript. With your help we have managed to resolve several inconsistencies in the manuscript and clear out many critical points, which hindered readability. We have made every effort to address all remarks. We hope you will recognize the extent of our efforts and find our responses satisfactory.

Thank you very much again for your review.

The Authors

**REPLIES TO THE COMMENTS**

**The reviewer's text is shown black, or replies red and planned changes red-italic. First we address the general comments and then we give replies to specific comments. We are also attaching a Supplement that will be added to a revised manuscript, and also address some of the comments.**

**Note that proposed changes are still subject to small changes, and not all citations are given!**

This is a very interesting paper presenting a modeling approach aimed at showing how the coupling between dissolution, transport in fracture formations, and horizontal head forms a low-karstified rock-blocks which prevent water seepage, up to a threshold of head difference between reservoirs. I very much like the approach, the results, and the layout of the article, and I can see how it will be well suited to HESS. However, three aspects require attention in this study:

**General Comment 1:** The authors heavily rely on jargon and assume that specific terms and mechanisms are known to the reader. This limits the impact of this study as it targets a narrow readership while overlooking HESS's broad readership. I outlined a few examples and ways to rectify this aspect in the detailed review below.

**Reply:** We fully agree with your suggestions. We have revised the introduction to better address a broader audience, including readers who are not specialists in karst science. We have emphasized the broader significance of karst aquifers and clarified the relevance of specific terms by replacing them with more familiar language and simpler, more accessible sentences.

The revised introduction (replacing **lines 1-30** in original manuscript) now reads as:

*Karst areas occupy approximately 15% of Earth's ice-free land surface. In these regions, the chemical erosion of rock by surface water and groundwater—known as karstification—produces distinctive*

*surface and subsurface features. Karst aquifers are among the most productive freshwater resources, supplying drinking water to around 20% of the world's population—and more than 50% in some countries (Ford and Williams, 2007). These aquifers are characterized by networks of solution conduits formed within initially fractured and porous rock. The development of these conduits, a process known as speleogenesis, is complex and spans nearly the entire period during which the rock is exposed to groundwater circulation (Dreybrodt, 1996; Yuan et al., 1993).*

*Karst aquifers can be conceptualized as hierarchically organized conduit systems embedded within a fractured porous matrix. These conduits serve as high-transmissivity flow pathways, channelling water from points of recharge to karst springs. Karstified rocks are also widespread in young orogenic belts, where the high-relief terrain provides favourable elevation differences that facilitate the construction of hydraulic structures such as dams and reservoirs (Milanović, 2000; Yuan et al., 1993; Shen et al., 1997).*

*However, the construction hydraulic structures in karst areas poses significant engineering challenges. The position, size, and connectivity of existing networks of solution conduits are extremely difficult to predict or detect. As a result, the extent of initial leakage from a reservoir is also hard to estimate (Milanović, 2000, 2018).  Furthermore, the high hydraulic gradients induced by such structures can accelerate the formation of new conductive pathways, potentially leading to a rapid increase in leakage—sometimes reaching intolerable levels within the operational lifespan of the structure (Dreybrodt, 1996; Gabrovšek and Dreybrodt, 2001, 2010; Romanov et al., 2003, 2007).*

**General Comment 2:** This work lacks some crucial aspects of the model. This complex model involves multiple processes over extended spatial and temporal scales, yet fundamental elements of this model are absent. What is the grid size and layout, and what is its sensitivity? Was there a convergence test for it? Was the model's sensitivity to key parameters assessed? The authors mention the stability of the solution but provide no data on it. The modeling aspect requires more than just the equations

used and their sequence; a clear section in the paper or an appendix should present these essential components of the model.

**Reply:** To address this comment, we have added a Supplement (S1, attached to this file), which introduces the fundamental elements of the model. Furthermore, the aspects of this comment (model layout, convergence criteria and stability) are addressed in replies to **specific comments 8, 9, 15 and 16**. We have also checked the equations and their sequence and revised them carefully, as shown in the reply to the **specific comment 12**.

**General comment 3:** The authors take an engineering approach to the results of their model, which is noteworthy as they draw a specific and tangible conclusion from it. However, the coupling between the transport and reactive mechanisms is a fundamental nucleation phenomenon, where the boundaries lead to an anisotropic, directional change. This phenomenon is observed in other experimental and numerical dissolution processes, where fingers or preferential flows emerge by this coupling (partial list: (Detwiler & Rajaram, 2007; Dijk et al., 2002; Edery et al., 2021; Kang et al., 2003; Molins et al., 2014; Nogues et al., 2013; Rege & Fogler, 1989; Shavelzon & Edery, 2022; Singurindy & Berkowitz, 2003)). Yet this is not referred to in this work, limiting its potential to draw a more general audience and a more general conclusion. These three aspects are echoed in the specific comments below, and I believe that while the last aspect may improve the paper, the first two are essential.

**Reply:** Thank you for pointing out an issue and providing interesting set of works. We have known some, but not all of them. We have extended the discussion to show this work in the context of nucleation phenomena and added some of the suggested references (see the response and suggested changes in reply to **Specific Comment 23**).

**Replies to specific comments**

1. The authors make an excellent case supporting their approach in the introduction, yet it will be beneficial to relate each aspect that is either added or missing in previous studies to the figure 1 illustration, thus providing a conceptual picture of the processes at hand.

   **Reply:** We have revised Figure 1 and it caption. This now relates the existence of stagnant cone in the water divide region within porous or fractured media. Lower flow rates in the water divide areas result in less dissolution and lower degree of karstification. The revised Figure 1 is shown below.

[Figure]

**Figure 1:** (a) The flow solution reveals the presence of a stagnant zone near the groundwater divide within the porous aquifer. (b) Karstification increases permeability and progressively lowers the water table over time, leading to the formation of blocks that are highly karstified and low karstified blocks (LKB). (c) When reservoirs are constructed, LKB can effectively obstruct leakage across the aquifer.

2. Line 22: "One of the primary issues is the intensification of the natural karstification process due to artificial hydraulic gradients, which can result in persistent and uncontrollable leakage throughout the structure's lifespan".

   This sentence exemplifies the narrow focus of the paper. In the second sentence of the introduction, we encounter jargon that is not properly explained. We have no idea what the artificial part is in these hydraulic gradients and why it leads to "persistent and uncontrollable leakage." I suspect that not all potential readers remember exactly what the "karstification process" is. HESS aims at a broad readership; therefore, an effort should be made to address this broad readership by thoroughly explaining the terms and concepts.

   **Reply:** We have addressed the issues raised in the comment in the new Introduction (Page 3).

3. Introducing Figure 1 earlier and including additional features, such as the permeability linked to fracture aperture changes and the chemical gradients that shape the LKB zone, can help rectify this. The latter will significantly aid in clarifying the cause-and-effect relationship that the authors seek to establish in their work. This approach will give readers a conceptual framework from the outset and elucidate the terms and concepts of the study.

   **Reply:** Figure 1 has been updated. See the reply to Comment 1 (Page 5).

4. Line 69: Explain what epikarst is.

**Reply:** Epikarst is the uppermost layer of vadose zone, an interface between the surface unconsolidated material and karstified carbonate rocks. Epikarst is capable of delaying, storing and locally rerouting vertical infiltration to the deeper zones of underlying karst aquifer. We have explained this specific term at its first appearance in the revised manuscript.

However, since the specific role of epikarst is not considered in this work, we have revised the paragraph (lines 70 to 75 of original manuscript), which now reads:

*As demonstrated by the models of Gabrovšek and Dreybrodt (2001) and Kaufmann (2003), the evolution of unconfined karst aquifers is most pronounced near the water table. In brief, these models do not explicitly account for the details of flow and dissolution in the vadose zone—the region between the surface and the water table—but instead assume gravitational flow that reaches a specific level of chemical saturation with respect to calcite. Nevertheless, within the phreatic zone, water is typically least saturated with respect to calcite near the water table, and thus possesses the highest potential for dissolution (often referred as aggressivity) in this region. The numerical methods employed in these studies primarily concentrate on the evolution of relatively homogeneous fracture networks, rather than the random fractures found in natural karst aquifers.*

5. Line 71: Not sure "aggressive" is clear in this context.

   **Reply:** See the change of the paragraph given above. Aggressive means that "the solution is capable of dissolving carbonates". We avoided the use of this term and used: *water is typically least saturated with respect to calcite near the water table, and thus possesses the highest potential for dissolution.*

6. Line 87: Explain what "speleogenesis" is.

**Reply:** We have introduced the term in the revised Introduction (see reply to General comment 1)m where we state: *The development of these conduits, **a process known as speleogenesis**, is complex and spans nearly the entire period during which the rock is exposed to groundwater circulation.*

7. Line 95: add a space after "(4)"

   **Reply:** Done.

8. While the criteria for transitioning between equations 1 and 2 are clear, the actual continuous transition between equations 1 and 2 or vice versa is not clear. How do we continuously transition from the laminar to the turbulent approximation without discontinuities in the flux? The flux mass balance presented in Eq. 4 should also be addressed in this context.
   Addendum: at the end of the paper, we learn that equation 2 was not used throughout the simulations (or so I understood from the following sentence: "Turbulent flow did not occur throughout the simulations initiated with natural, original fractures."). If that is the case, why present it?

   **Reply:** The model allows turbulent flow, however as noticed by the reviewer, in the presented simulations turbulence never occurred. Transition from laminar to turbulent flow and vice versa is a complex phenomenon. Most models calculate friction factors from the Reynolds number, which is also criterion for onset of turbulence. Different approaches of how to deal with noncontinuous phenomena are being used, modellers often invoke some type of hysteresis, and another elegant approach is use of Churchill equation, which gives a smooth continuous friction factor as a function of Reynolds number in the transition zone. How precise we want to model, depends on what we want to model and how important the proper treatment of laminar/turbulent transition is. Experiences from previous modelling of karst networks evolution showed that different

algorithms gave practically same results. Therefore, here we use a rather crude approach, where we simply take the flow with higher frictional losses. In fact this means that we neglect transition zone and switch directly from laminar to fully turbulent flow. Figure R1 shows flow through a fracture as a function of hydraulic gradient calculated for laminar flow and from Lomize approximation. For Re<2020 Eq.1 is used and Eq.2 is used for Re>2020.

We have added a text at the end of section 2.1 (before the Line 110 in the original text):

*The transition from the laminar to the turbulent flow regime is here introduced at the point where turbulent frictional losses exceed those associated with laminar flow. The factor in Lomize equation (Eq. 3), is set so that transition occurs at a Reynolds number of 2020. This simplification omits the transitional flow zone, resulting in a smooth increase in flow rates as turbulence sets in. However, in the results presented, the flow remains within the laminar regime throughout the simulations.*

[Figure]

Figure R1. The approximation of transition between laminar and turbulent flow.

9. Line 121: Is Figure 2 an illustration, actual layout, or sub-layout of the fractured domain? Also, the dimensions are critical in understanding the model framework (Reynolds number, head differences, etc.). Referring to boundary, seepage, and head without their dimensions seems inadequate.

**Reply:** Figure 3 (renumbered) presents just the concept of how the position of WT is calculated. We have extended the caption to make it more readable. Furthermore, since both reviewers found it disturbing, we have dropped the somehow misleading terminology of inner, middle, and outer iteration and changed it to **Confined Flow Calculation, node-by-node iteration, layer-by-layer iteration. The notation is changed throughout the test.**

[Figure]

*Figure 3.(renumbered) Conceptual presentation of flow calculation in the fracture network. (a) Boundary conditions for Confined flow calculation performed at **every** iterative step. (b) Node-by-node iteration: testing wetting of a dry node across a layer. (c) Layer-by-layer iteration. $H_i$ and $H_j$ are heads at wetted nodes, $q_w$ is direct recharge from the vadose flow, z is elevation of the node.*

10. Section 2.2.2: Please refer to Figure 2 when explaining the necessary steps in finding the water table.

**Reply:** Thank you for pointing to insufficient referencing to Figure 2. In the revised manuscript we have updated the text to fit the figure.

*Introductory* paragraph (Line 115) of Section 2.2 now outlines the search for the water table; it now reads:

*Each evolution time step requires the calculation of flow, dissolution, solute transport, and the corresponding changes in fracture aperture. Accurate flow calculation depends on determining the position of the water table at each time step. This is achieved through an iterative process, in which the water table position is updated until specific convergence criteria are met, as described in Section 2.2.2. The procedure is illustrated in Figure 2. At each iteration, the current approximation of the water table defines a set of boundary conditions (Figure 2a): prescribed recharge at water table nodes, seepage conditions (where hydraulic head equals surface elevation) at seepage nodes, and either no-flow or constant-head conditions elsewhere. The flow solver (Confined flow solution) is then invoked to compute flow rates and hydraulic heads at all nodes. Water table nodes are subsequently searched node by node within each layer (Figure 2b, node-by-node iteration) and layer-by-layer throughout the domain (Figure 2c, layer-by-layer iteration).*

*We also intend to add a flow chart (Figure R2), which will summarise the steps described in section 2.2.2.*

[Figure]

Figure R2: Flowchart of the flow solution.

11. Section 2.3: The "$H_2O$-$CO_2$-$CaCO_3$" system is heavily influenced by pH. Although the $CO_2$ concentration can provide an approximation, this is only valid within a limited range of pH values. Please address this aspect.

**Reply:** In most carbonate karst areas, the $CO_2$ is considered as a primary source for the dissolution potential of natural water with respect to carbonates. The pH is of course a master variable of the system and is as such used to calculate equilibrium. In a pure $CO_2$-water-calcite system, the protons are primarily delivered from the hydration of $CO_2$ molecules, and consumed by the

formation of bicarbonate. Therefore, the availability of $CO_2$ in this system, determines the solubility, or equilibrium concentration. The undersaturation defines the rate of dissolution according to the experimentally derived rate laws (Eq. 7 and 8).

We have revised paragraph (from line 158) to better clarify these concepts. It now reads:

*where $k_1$ and $k_n$ are rate constants (in $mol·cm^{-2}·s^{-1}$), C represents the concentration of $Ca^{2+}$ ions (in $mol·L^{-1}$), and $C_{eq}$ denotes the equilibrium concentration of the $H_2O$-$CO_2$-$CaCO_3$ system. The $C_{eq}$ can be calculated from the concentration of calcium and dissolved $CO_2$ of the solution at the water table. This equilibrium state results from complex flow and dissolution processes occurring in the vadose zone, which are beyond the scope of this study. We take uniform equilibrium concentration (2 mmol/L) and uniform saturation ratio ($C_{in}=0.92C_{eq}$) at water table nodes. The reaction follows a linear rate law up to the switch concentration $C_s$, transitioning to a nonlinear rate law between $C_s$ and $C_{eq}$. Here we assume that rates are entirely controlled by surface reaction, and ignore the concentration gradient perpendicular to flow. This approximation is valid for situations where the solutions are close to equilibrium, which is mostly the case in the presented scenarios. The reaction order n and the switch concentration....*

12. Furthermore, a convective approach for the laminar case must approximate the reactant depletion as it reaches the fracture surface by some rate law. Since eq. 9-11 do not consider the fracture thickness b_{ij} while the illustration in figure 3 does, I am a bit confused about this matter. Please provide clarity on this aspect.

**Reply:** We agree that this can be told better. The aperture controls the flow rates, which in turn control the increase of calcium concentration along the fracture. Furthermore, the dissolution rate depends on the surface reaction, determined by concentration of species at the solution-mineral interface, and diffusion transport, which is controlled by concentration gradients within

the fracture. Diffusion rates, among others depend on aperture width, which is not considered in the model. We here approximate the dissolution rates by experimentally determined surface reaction rates as given in Eq. 7 and 8. Taking the fact that the inflowing solution is close to equilibrium, the approximation of surface controlled reaction is reasonable.

We have clarified this in the text. Furthermore, we have moved the calculation of change of concentration within a fracture and corresponding mass and aperture change to appendix.

The paragraph (Lines 166-185) will be replaced by:

*The change in concentration along an individual fracture can be determined by applying the principle of mass conservation within a water parcel passing along the fracture. This can be easily analytically calculated for parallel plan (the flow perimeter P(x) is constant) fracture, as shown in the supplements. Change of concentration along the fracture can be converted to mass removed from the fracture walls and the change of aperture during time Δt,*

$$\Delta b = \frac{\Delta C \; q \; M_{CaCO_3}}{\rho \; L} \Delta t \qquad (9)$$

*where L is the fracture length, ρ is the density of calcite (2.5 g/cm$^3$) and $M_{CaCO_3}$ is the molar mass of calcite (100 g/mol), and q the linear flow density.*

13. Line 182: Is the mass removal term converted to a volume change from which the b_{ij} is updated? If so, what is the conversion constant from moles to volume?

    **Reply:** This is now described in the paragraph above.

14. Line 204: Correct grammar in "These analyses"

    **Reply:** Done.

15. Figure 4: While we agree on the analytical solution deviation, the magnitude of the deviation between the numerical value and the analytical solution is around 50%, which necessitates further elaboration on why it is so large.

**Reply:** We fully agree that the analytical solution presented is not sufficient for the flow validation. Actually the analytical solution is based on Dupuit approximation, which neglects vertical flow and seepage face. We have now used MODFLOW and its DRAIN module to test the validity of our solution on basic cases and moved the test results to Supplements.

[Figure]

Figure S2.1. Water tables and corresponding heads validation with MODFLOW and Dupuit analytical models for two distributed uniform recharge conditions.

16. Section 2.4.2: It's hard to understand how well the model performs since we do not have any specific case to compare it to, aside from the analytical solution, which we established as inadequate. How can we be sure that the model works as it should? In terms of numerical analysis, there is no mention of grid size sensitivity, convergence, or stability of the numerical code, which are standard practices. The authors should present these aspects of the code so that we can appreciate it accordingly.

**Reply:** As written above, the position of the water table estimation is in a good agreement with with MODFLOW calculation for porous media with equivalent hydraulic conductivity. The accuracy of confined flow solution is assured by flow balance and head tolerance ($H_{tol} = 10^{-4}$ m and $Q_{tol} =$ 0.1 cm$^2$/s). The flow solver uses preconditioned conjugate gradient approach for sparse matrix.

Although we deal with *irregular grid*, the accuracy also somehow depends on the fracture length. Namely, the head at the water table nodes is between the elevation of the WT node and the elevation of the nearest dry neighbour node. The widening of the fracture connecting these nodes is not considered. This inconsistency is smaller for small grids.

17. Line 218: "on the both sides" should be "on both sides"

**Reply:** Done.

18. Line 219: "…the algorithm performed well in modelling the water table in heterogeneous network." Well, compared to what? These statements appear throughout the paper, yet they are not supported by any comparison or measurement that helps us understand what "well" means. The only reference is to the ill-fitted analytical solution.

**Reply:** We hope that we have adequately responded to the comment in answers given above.

19. Table 1: What are the dimensions of the directional term? Angle?

**Reply:** Yes, this refers to the angle between the fracture and the horizontal, expressed in degrees. We have corrected the directional reference of the angle in the revised manuscript.

20. Additionally, the mean length of the fractures is quite large. Is this realistic? Why was this length chosen? It appears that ten well-connected fractures may dominate the simulation

**Reply:** The choice of parameters for stochastic fracture generation is rather arbitrary, but fracture lengths span quite a large interval which corresponds to possible natural settings. The realisation used here is just one of the possible ones, used to test the concept of LKB. Further modelling would be needed to show the extent and hydraulic contrast of LKB in various setting, which remains a further perspective.

21. Line 252: Please clarify the term "evolution time step."

**Reply:** We have corrected "evolution time step" to "time step" in the revised manuscript. The system goes through a set of stationary states, when flow dissolution and widening is calculated. The time spent in the state is called a time step. The choice of time step is heuristic, and is a compromise between the accuracy and computational demand. We have compared the result of simulation for different time steps and the differences become negligible, when time steps are smaller than the one taken in this work.

22. Line 255, figure 8 caption: What does "ka" stand for? I'm assuming it refers to time, but no dimensions have been provided. This should be clarified in Figure 8, not Figure 10. Additionally, the heat map for the aperture could be confused with the head heat map. It is unclear whether it is necessary.

**Reply:** The "ka" stands for one thousand year. We have corrected it with an explanation in Figure 8. To avoid confusion, we have also rotated the heat map of aperture Figure 9 to be

vertical in the revised manuscript. Figure 8 and Figure 9 have been renumbered to Figure 6 and Figure 7. Figure 7 is shown below:

[Figure]

**Figure 7(renumbered)** Evolution of fracture aperture at 0 ka (a), 10 ka (b) and 20 ka (c) under natural conditions.

23. Section 3.3: In this section, the authors relate the change in aperture to the location of the aquifer, as evident in Figure 11, and also relate the change in flux in a similar manner. However, the cause and effect suggest that the higher potential near the boundaries dictates higher fluxes, and as the flux increases, so does the reaction rate, which widens the aperture. This is a nucleation phenomenon observed in many studies on dissolution, specifically in the context of permeable structures.

**Reply:** We agree that the text in section 3.3 needs some revision to more clearly address the feedback mechanisms between flow and dissolution rates.

We have added the paragraph at the end of Section, which discusses the issues raised in the comment:

*Karstification represents a form of nucleation, where flow-induced dissolution and changes in porosity are coupled through feedback mechanisms* (Eder et al., 2021; Molins et al., 2014) . *In unconfined aquifers under constant recharge conditions, dissolution is most intense near the water table. This process creates a highly permeable fringe that effectively channels inflow toward both sides of the water divide. As this fringe migrates downward across the aquifer cross-section, it leaves behind a distinctive porosity imprint. Simultaneously, it inhibits deeper penetration of the inflowing solution, favoring the preferential development of horizontal fractures. Moreover, flow along the water table increases progressively from the water divide toward the discharge points. As a result the water divide zone is less karstified than the regions close to the output. Similar anisotropic, directional changes, including fingers or preferential flows, has also been observed through experimental studies and other numerical simulations (Shavelzon and Edery, 2022; Singurindy and Berkowitz, 2003).*

24. Line 295: I find the K calculation very interesting. To begin with, the fact that there is only dissolution in this setup means that the LKB is a "residual" permeability, indicating that while some permeabilities have increased, the LKB remained unchanged. However, the K is calculated directly for a subsection, and figure 12, as well as a close examination of figure 8, shows that there is an anisotropic change in the aperture, where horizontal fractures experience more dissolution than the vertical fractures. This also leads to the formation of the LKB and the observed changes in flux. However, this structural anisotropy is not discussed or quantified in this context, although it is clear that the horizontal head difference drives the anisotropy. This emergence of anisotropy can be found in many studies on rock dissolution, where preferential flows arise due to these boundary conditions coupled with reactive transport. As this emergent behaviour appears in similar fields, the connection should be made among them.

**Reply:** We have included discussion in a previous comment.

Reference list:

Detwiler, L. R., and Rajaram, H.: Predicting dissolution patterns in variable aperture fractures: Evaluation of an enhanced depth-averaged computational model, Water Resour. Res., 43, W04403, https://doi.org/10.1029/2006WR005147, 2007.

Dijk, P. E., Berkowitz, B., and Yechieli, Y.: Measurement and analysis of dissolution patterns in rock fractures, Water Resour. Res., 38, https://doi.org/10.1029/2001WR000246, 2002.

Edery, Y., Stolar, M., Porta, G., and Guadagnini, A.: Feedback mechanisms between precipitation and dissolution reactions across randomly heterogeneous conductivity fields, Hydrol. Earth Syst. Sci., 25, 5905–5915, https://doi.org/10.5194/hess-25-5905-2021, 2021.

Kang, Q., Zhang, D., and Chen, S.: Simulation of dissolution and precipitation in porous media, J. Geophys. Res., 108, https://doi.org/10.1029/2003JB002504, 2003.

Molins, S., Trebotich, D., Yang, L., Ajo-Franklin, J. B., Ligocki, T. J., Shen, C., and Steefel, C. I.: Pore-scale controls on calcite dissolution rates from flow-through laboratory and numerical experiments, Environ. Sci. Technol., 48, 7453–7460, https://doi.org/10.1021/es5013438, 2014.

Nogues, J. P., Fitts, J. P., Celia, M. A., and Peters, C. A.: Permeability evolution due to dissolution and precipitation of carbonates using reactive transport modeling in pore networks, Water Resour. Res., 49, 6006–6021, https://doi.org/10.1002/wrcr.20486, 2013.

Rege, S. D., and Fogler, H. S.: Competition among flow, dissolution, and precipitation in porous media, AIChE J., 35, 1177 – 1185, https://doi.org/10.1002/aic.690350713, 1989.

Shavelzon, E., and Edery, Y.: Modeling of Reactive Transport in Porous Rock: Influence of Peclet Number, EGU22-8059, https://doi.org/10.5194/egusphere-egu22-8059, 2022.

Singurindy, O., and Berkowitz, B.: Evolution of hydraulic conductivity by precipitation and dissolution in carbonate rock, Water Resour. Res., 39, W1016, https://doi.org/10.1029/2001WR001055, 2003.

**Reply:** Thank you for your suggestion. We have reviewed the recommended papers and found that they offer valuable new perspectives on our work. While we have not yet had the opportunity to study them all in depth, we have cited several in the revised manuscript (see above). We intend to incorporate these insights more fully into our future research.

**Supplements**

**S1 Model details & validation**

At each time step, the convergence status is recorded in the runtime output file. An example is available on our GitHub repository at Fracturetokarst2024/slurm-17392021.out.. This file includes key indicators such as the number of iterations required for solving the iterative flow equations for confined flow solver, water head errors, and total flux balance. Iterations are controlled internally both node-by-node and layer-by-layer. If the confined flow solver encounters convergence issues, the model reverts to the last successful state and proceeds by testing an the next dry node. To assess the model's accuracy, results were benchmarked against MODFLOW simulations, as described in the subsequent section.

We also explored the model's behavior under different flow regimes by simulating the coexistence of turbulent and laminar flows. These tests confirmed the numerical stability of the model. Additionally, the model is capable of generating random fractures, which are used directly in flow and dissolution computations. These fractures vary in shape and size and are not constrained by the underlying computational grid.

A high-performance computing environment is essential for achieving long-term stability in water table simulations. The calculations were performed on a platform using a Chinese Hygon C86 7185 32-core processor, running CentOS Linux 7. For these simulations, we utilized 8 cores on a single compute node.

**S 2 Test of the water table in a homogeneous fractured aquifer**

To verify the numerical model, we first compare the results for the homogeneous network with MODFLOW (Harbaugh et al., 2000) and the analytical solution derived using the Dupuit assumption, which can be expressed as follows:

$$H = \sqrt{H_0^2 - \frac{w}{K_x}X^2 + \frac{w\,S}{K_x}X} \tag{S1}$$

where $H_0$ is the river base level, $K_x$ is the equivalent horizontal conductivity in m/d, $S$ is the aquifer length, $X$ is the distance from the left river boundary, and $w$ is the intensity of rainfall recharge.

The homogeneous fracture network is shown in Figure S2.1. We assume translational symmetry and therefore use 2D domain populated by fracture. The horizontal dimension of the domain is 1000 m and vertical 400 m. The distance between the fractures in both the X direction and the Y direction is 10 m. The aperture is 0.01 cm for all the fractures. Along the two side boundaries, nodes lower than 200 m in height are given a constant water head of 200 m. Nodes above 200 m have seepage boundary conditions. The two recharge conditions were tested at 400 mm/a and 800 mm/a.

Within the MODFLOW validation, we calculate the equivalent horizontal $K_x$ and vertical $K_y$ by treating the aquifer as having confined water head boundaries in the X and Y directions, respectively. The conductivities are proportional to the ratio between the resulting flux and the head difference. The horizontal and vertical K values are virtually identical, both approximately 0.00705 m/d. The DRAIN module in MODFLOW was used to model the seepage face by setting each boundary node above the constant head with drainage function, which worked as its water head became higher than the drain's elevation. In Figure S2.1, water table nodes are labeled blue and the corresponding heads have orange labels. The water tables obtained via the Dupuit assumption are always lower than the simulated water table since the vertical flow and seepage face are not considered. Note that the heads at water table nodes are higher than their elevation, but below the elevation of the nearest dry node above them. The water tables simulated with our method are nearly the same as the MODFLOW simulation. Additionally, we can see the seepage face boundaries on both sides, which evidently do not exist in the analytical solution. The analysis demonstrates the effectiveness of our algorithm.

[Figure]

**Figure S2.1:** Water tables and corresponding heads validation with MODFLOW and Dupuit analytical models for two distributed uniform recharge conditions.

**S3 Modeling a water table in a heterogeneous fractured aquifer**

The next step is to test the solution for heterogeneous network. We use the same setting as for the homogenous network but with random generation of fractures and two recharge conditions, 200 mm/a and 400 mm/a. The equivalent horizontal and vertical K values are 0.00469 m/d and 0.00434 m/d, respectively. The number of outer iterations in all evolving time steps varied from 5 to 35. The process of searching for a water table takes approximately 3 to 4 hours during the initial modeling stages of karst evolution, and it is performed on a high-performance computing platform that utilizes 8 cores.

The fracture flow and water table data are shown in Figure S3.1 and Figure S3.2. The water table is discontinuous because of the inhomogeneous distribution of fractures. Only a few nodes for the simulated water table are lower than the analytical water table. The difference between the elevation and head at water table nodes varies due to the heterogeneity of the network. The seepage faces above the constant head boundaries on the both sides of the domain, are successfully simulated as the Signorini boundary (Jiang et al., 2013). Considering these two recharge conditions, the algorithm performed well in modelling the water table in heterogeneous network.

[Figure]

**Figure S3.1.** Modelling the phreatic flow in random fractures under 400mm/a and 200mm/a distributed uniform recharge conditions.

[Figure]

**Figure S3.2:** The simulated and analytical results of random fracture water tables and water heads of corresponding nodes.

**S4 Changes of concentration within individual fracture and on a network scale**

S4.1 Dissolution and change of concentration along an individual fracture

[Figure]

**Figure S4.1:** Calculating Ca²⁺ concentration along one single fracture and at fractures joint node and near the water table.

To calculate change of concentration within a single fracture, we use a Lagrangian approach and imagine a water parcel with volume $dV=Pdx$ moving along the fracture with velocity $v=Q/A$, where A is a flow crosssection, and calculate the change of concentration within the parcel (Figure S4.1). In a time $dt$ the change of concentration in a parcel is equal to:

$$dC = F(c) \cdot P \cdot dx \cdot dt/dV \tag{S2}$$

Where $P$ is flow perimeter and $Pdx$ is the surface area of water/rock contact. Rearranging the equation gives:

$$\int_{C_i}^{c(x)} \frac{dc}{F(c)} = \frac{P}{Q}x = \frac{2(b+w)}{qw}x \approx \frac{2}{q}x \tag{S3}$$

The last term is an approximation for a wide fracture with the lateral width $w$ and aperture $b$, where $w>>b$; $q$ is flow rate per unit width. Using rate Equations (Eq. 7 and Eq. 8) for $F(c)$, we get:

$$C(x) = C_{eq} - \left(C_{eq} - C_i\right)e^{-\left(\frac{2k_1}{q\,C_{eq}}x\right)}, \quad (C < C_s) \tag{S4}$$

$$C(x) = C_{eq} - C_{eq}\left(C_{eq} - C_i\right)\sqrt[3]{\frac{q\,C_{eq}}{\left(C_{eq} - C_i\right)^3 6k_4x + q\,C_{eq}^4}}, (C_s < C < C_{eq}) \tag{S5}$$

Where $k_1$ and $k_4$ are rate constants. $C_{eq}$ and $C_s$ are the equilibrium concentration and the switch concentration of Ca²⁺ ions.

The change of concentration $\Delta C$ at the outlet of the fracture is given by:

$$\Delta C = C_{eq} - C_{eq}e^{-\left(\frac{2k_1}{q\,C_{eq}}L\right)} + C_i\left(e^{-\left(\frac{2k_1}{q\,C_{eq}}L\right)} - 1\right), (C_i < C_s, x_s > L) \tag{S6}$$

$$\Delta C = C_{eq} - C_i - 0.1\,C_{eq}^2\sqrt[3]{\frac{q\,C_{eq}}{\left(0.1C_{eq}\right)^3 6k_4L + q\,C_{eq}^4}}, (C_i < C_s, x_s < L) \tag{S7}$$

$$\Delta C = C_{eq} - C_i - C_{eq}^2\sqrt[3]{\frac{q\,C_{eq}}{\left(C_{eq} - C_i\right)^3 6k_4L + q\,C_{eq}^4}} + C_{eq}C_i\sqrt[3]{\frac{q\,C_{eq}}{\left(C_{eq} - C_i\right)^3 6k_4L + q\,C_{eq}^4}}, (C_i > C_s) \tag{S8}$$

Where $x_s$ is the switch distance of $C_s$. If $C_i < C_s$ and $x_s > L$, dissolved mass is calculated from Eq. (S6). If $C_i < C_s$ and $x_s < L$, Eq. (S7) is used. If $C_i > C_s$, Eq. (S8) is used for dissolved mass directly.

S4.2 Following concentration at the network scale

To assure that concentrations at the input nodes are always known, we follow the procedure of Siemers and Dreybrodt (1998) and Gabrovšek and Dreybrodt (2000). The process begins at the network's boundary nodes with the highest hydraulic heads, where head or flux values and concentrations are prescribed. Calculations then proceed sequentially along the hydraulic gradient. As illustrated in Figure S4.1(b and c), the concentration $C_j$ at the node $j$ is calculated using the complete mixing assumption. This involves computing the flow-weighted average of the incoming concentrations:

$$c_j = \frac{\sum_i C_{ij}^{out} q_{ij} + \sum_k C_0 q_{wk}}{\sum_i q_{ij} + \sum_k q_{wk}} \tag{S9}$$

Where $q_{wk}$ and $C_0$ are flow and concentration of direct recharge at water table nodes; $q_{ij}$ and $C_{ij}{}^{out}$ are the output flow and concentration of fractures connecting nodes $i$ and $j$; $i$ sums over confined nodes that deliver flow to j, and k runs over direct input at the water table.

**References**

Harbaugh, A. W., Banta, E. R., Hill, M. C., and McDonald, M. G.: MODFLOW-2000, the US Geological Survey modular ground-water model: User guide to modularization concepts and the ground-water flow process, U.S. GEOLOGICAL SURVEY, Open-File Report 00-92, https://doi.org/10.3133/ofr200092, 2000.

Jiang, Q., Yao, C., Ye, Z., and Zhou, C.: Seepage flow with free surface in fracture networks, Water Resour. Res., 49, 176-186, https://doi.org/10.1029/2012WR011991, 2013.

---

## Author Comment (AC2)

**REPLY TO THE REVIEWER'S COMMENTS**

We sincerely appreciate your thorough and thoughtful review. Many of the comments and suggestions required careful consideration and prompted substantial revisions to the manuscript. With your help we have managed to resolve several inconsistencies in the manuscript and clear out many critical points, that hindered readability. We have made every effort to address all remarks. We hope you will recognize the extent of our efforts and find our responses satisfactory.

Thank you very much again for your review.

The Authors

**The reviewer's text is shown black, or replies red and planned changes red-italic. First we address the general comments and then we give replies to specific comments. We are also attaching a Supplement that will be added to a revised manuscript, and also address some of the comments.**

**Note that proposed changes are still subject to small changes, and not all citations are given!**

The authors present an interesting methodology to model karst evolution due to flow, transport and the dissolution within a fracture network, below a fluctuating water table. Although the approach is mainly presented within a limited context of dams and reservoirs, such modeling approach could be useful in a much wider context as well. I think the paper could be a valuable contribution to the journal, it is well written and easy to follow. I have a few recommendations that would improve the text:

**General comments:**

1. While the individual steps of the modeling are mostly well presented, I would find useful to have an overview of the whole modeling process. In my opinion a flowchart would greatly help the methodology section. One thing I found confusing is  the mixing of terms inner-outer iterations, steps - it was difficult to follow which steps happen within a single timestep etc. Using a flowchart could easily alleviate this issue, and would give a good entry point to the modelling process.

**Reply:** We acknowledge that the terminology was rather confusing, stemming from a bias inherent in the research process. To address this, we have abandoned the confusing notation for iteration levels. Instead, we now employ the terms ***confined flow solution, node-to-node iteration, and layer-by-layer***

*iteration* to more accurately describe the computational procedures. Additionally, we have included a flowchart Figure R1 that provides a comprehensive overview of the entire simulation workflow.

[Figure]

*Figure 3.(renumbered) Conceptual presentation of flow calculation in the fracture network. (a) Boundary conditions for Confined flow calculation performed at **every** iterative step. (b) Node by node iteration: testing wetting of a dry node across a layer. (c) Layer by layer iteration. $H_i$ and $H_j$ are heads at wetted nodes, $q_w$ is direct recharge from the vadose flow, z is elevation of the node.*

[Figure]

*Figure R1.* *The flowchart of the whole modelling process. At each time step, the new position of the water table is determined through an iterative process. Subsequently, the coupled flow, dissolution, and transport equations are solved at both the individual fracture scale and the fracture network scale. Based on these results, the fracture apertures are updated accordingly. The modified fracture network then serves as the basis for calculations in the following time step.*

2. How would you validate such modeling approach? Do you see a potential to compare the results with field measurements? Or did you consider validating the individual modelled processes against other modeling approaches? I think such validation would be an important step for presenting such new methodology. It would also be interesting to see, how the modeling results compare against other methods (such as equivalent porous media, or a static DFN model).

**Reply:** We agree that the analytical solution presented is not sufficient for the flow validation. Actually the analytical solution is based on Dupuit approximation, which neglects vertical flow and seepage face. We have now used MODFLOW and its DRAIN module to test the validity of our solution on some basic cases and moved the test results to Supplements.

[Figure]

**Figure S2.** Water tables and corresponding heads validation with MODFLOW and Dupuit analytical approximation for two distributed uniform recharge conditions.

The accuracy of confined flow solution is assured by water head and flow balance tolerance ($H_{tol} = 10^{-4}$ m and $Q_{tol} = 0.1$ cm$^2$/s). The flow solver uses preconditioned conjugate gradient approach for sparse matrix.

3. In general, I think there are too many figures in the manuscript, and many of them could be moved to the supplements. Fewer figures with less information, would better highlight the interesting results from the modeling.

**Reply:** We agree with the suggestion. The model testing section (2.4.1) and the detailed calculations from Section 2.3 have been moved to the Supplements. In their place, we have provided a concise summary of the workflow without delving into technical details. Readers interested in the full methodology can refer to the appendix for more information. However, we would prefer to retain Figures 10–12 in the main manuscript, as they convey essential insights into the formation of the LKB.

**Specific comments**

1. L30: The concept of LKB is very important for this paper, but this explanation is too short for it. Please explain it better.

**Reply:** Thanks for your valuable comment. We have further explained the concept of LKB in the revised manuscript (Introduction from Line 30 of the OM), that will now read:

*Basic flow solutions in unconfined porous aquifers with constant recharge, suggest a relatively stagnant flow zone in water divide area (Rhoades and Sinacori, 1941; Tóth, 1962; Liang et al., 201). This also applies to fractured aquifer and can have important implication for karstification, where low flow zone may also result in less karstified zone. Such low karstified blocks have been recognised in water divide regions of real karst aquifers and proven to be effective in mitigation of leakage from reservoirs in karst areas (Yuan et al., 1993; Milanović, 2000; Xu and Yan, 2004). See Also Figure 1.*

We have also *updated* Figure 1 and its caption, to introduce the LKB concept in a clearer manner:

[Figure]

Figure 1. (a) The flow solution reveals the presence of a stagnant zone near the groundwater divide within the porous aquifer. (b) Karstification increases permeability and progressively lowers the water table over time, leading to the formation of blocks that are highly karstified and low karstified blocks (LKB). (c) When reservoirs are constructed, LKB can effectively obstruct leakage across the aquifer.

2. L35: "The pattern of groundwater flow in water divide areas also suggests the possibility of an LKB in karst aquifer." - why?

**Reply:** We hope that the response to previous comment also addresses this one.

3. L50 onwards: this is a very good motivation for the paper

**Reply:** We appreciate your feedback.

4. L90: all these steps happen within one time step

**Reply:** We apologize for the lack of clarity in the original presentation. We hope that the revised flow chart (Figure 2), along with the updated list in lines 91–95, now clarifies the intended workflow and structure.

5. L128: Does this mean you are aiming for a steady state within the iterations?

**Reply:** Yes. The system goes through a sequence of steady states. Within each time step, a stable solution for flow, dissolution and transport are found and change of fracture apertures are calculated within a time step $\Delta t$. We have made this more clear; see reply to the comment above.

6. L147: what does middle iteration mean?

**Reply:** Please refer to our response to General Comment 1 and the accompanying flowchart. We have revised the terminology and introduced a new flowchart to reflect these changes. See also reply to the Comment 7.

7. L149: a flowchart would be great here

Reply:  We have added a flowchart of the flow calculation to the supplements, as shown below. The text in section 2.2.1 and 2.2.2 will refer to it.

[Figure]

Figure R2. The flowchart of flow solution.

8. L150: What does layer-by-layer mean here? This part in general is quite confusing.

**Reply:** We apologize for the lack of clarity in the original submission. To address this, we have revised the caption in Figure 2 (now updated as Figure 3), introduced a new flowchart (Figure R2), added a supplementary diagram summarizing the iterative process of flow calculations (Figure R1) and changed the text correspondingly in Section 2.2.

9. L188: What does homogeneous mean here? Are you verifying against an equivalent porous media model?

**Reply:** Yes, "homogeneous" refers to a regular grid of fractures with a uniform aperture distribution. For such a configuration, it is straightforward to determine an equivalent porous medium with corresponding hydraulic conductivity for use in MODFLOW simulations.

10. L213: More details about the high-performance platform is needed (CPU type)

**Reply:** We have added the details about the computing platform to the Supplements S1, as shown below:

*"The CPU type is Chinese Hygon C86 7185 32-core Processor, and the operating system is CentOS Linux 7. We use 8 cores on a single node mainly for the long time stable calculation."*

11. L227: This section is unclear to me. Are we talking about in the dam or somewhere else? What is a karst reservoir in this context?

**Reply:** Thanks for your comment. The *reservoir* refers to a body of water behind the dam. We model the aquifer between A and B, e.g. between the reservoir on the left and the river in the right. The text will be changed clarify this aspect.

12. L235: This is a very good case study site for the approach.

**Reply:** Thanks for your comment. We appreciate your feedback.

13. Table 1: How did you choose the parameters?

**Reply:** We adopted parameters from several studies that include data from the Luojiaao interfluve aquifer. Observations of $CO_2$ and $Ca^{2+}$ concentrations were collected from three boreholes and two springs. It is important to emphasize that these data are used in an illustrative manner—serving only

to loosely relate the model to a field site. The Luojiaao interfluve is presented solely as a motivating example, without any intention to construct a fully realistic or site-specific model of the system.

14. How did you choose the simulation length?

**Reply:**  We simulated the evolution for 100,000 years, when the water table has dropped to the base level and no more dynamics is observed except for the conduit evolution at the base level. However, analysis has been made for evolution times up to 20,000 years, when the water table was in similar position as revealed by borehole data.

15. Figure 9: These plots a bit confusing, can you rotate them so the x-y plane is vertical?

**Reply:** We agree; sometimes authors need a kick from their bias. We have rotated the figure.

[Figure]

**Figure 7(renumbered).** Evolution of fracture aperture at 0 ka (a), 10 ka (b) and 20 ka (c) under natural conditions.

16. L272: "The water table with dissolution fringe mainly descends through the upper section, which experience the evident change in aperture" - This sentence is unclear

**Reply:** We agree that the sentence, as it currently stands, appears somewhat out of context. We will remove it and instead incorporate a new paragraph at the end of the section (starting from Line 285) that summarizes the underlying mechanism and conveys the intended message more clearly. The revised paragraph reads:

*Karstification represents a form of nucleation, where flow-induced dissolution and changes in porosity are coupled through feedback mechanisms (Eder et al., 2021; Molins et al., 2014). In unconfined aquifers under constant recharge conditions, dissolution is most intense near the water table. This process creates a highly permeable fringe that effectively channels inflow toward both sides of the water divide. As this fringe migrates downward across the aquifer cross-section, it leaves behind a distinctive porosity imprint. Simultaneously, it inhibits deeper penetration of the inflowing solution, favoring the preferential development of horizontal fractures. Moreover, flow along the water table increases progressively from the water divide toward the discharge points. As a result the water divide zone is less karstified than the regions close to the output. Similar anisotropic, directional changes, including fingers or preferential flows, has also been observed through experimental studies and other numerical simulations (Shavelzon and Edery, 2022; Singurindy and Berkowitz, 2003).*

17. Fig. 10-11-12: I see a lot of redundancy in these figures, are they all important? Consider moving some of them to the supplements.

**Reply:** We have considered avoiding the redundancy of the figures. Since we want to highlight the part of LKB formation, we would like to keep these figures. We have updated the text to refer to them.

18. L347: "Considering..." - elaborate this statement more

**Reply:** We have elaborated the statement. We have changed text from Line 344 to 350 that now reads:

*The results show moderate and acceptable increase of leakage within the expected life span of the dam (about 100 years). However, we have to be aware that model is idealisation of reality, and that further structural, speleological and hydrological data would be required to give a more reliable site-specific prediction.*

19. Where do you see here the link between the model and the real case? How could the model be used in this specific setting?

**Reply:** We believe that we have partially addressed this question in our previous response. However, to be sincere, real karst aquifers are extremely difficult to characterize with sufficient precision to make such predictions feasible. Nevertheless, the conceptual approach tested in this work provides valuable input for practical applications.

20. L366: this section title is unclear to me.

**Reply:** We have changed the title to "Concept of LKB in water divide region" We have also change title of section 5.2 to "Shortcomings of the model".

**Supplements**

**S1 Model details & validation**

At each time step, the convergence status is recorded in the runtime output file. An example is available on our GitHub repository at Fracturetokarst2024/slurm-17392021.out.. This file includes key indicators such as the number of iterations required for solving the iterative flow equations for confined flow solver, water head errors, and total flux balance. Iterations are controlled internally both node-by-node and layer-by-layer. If the confined flow solver encounters convergence issues, the model reverts to the last successful state and proceeds by testing an the next dry node. To assess the model's accuracy, results were benchmarked against MODFLOW simulations, as described in the subsequent section.

We also explored the model's behavior under different flow regimes by simulating the coexistence of turbulent and laminar flows. These tests confirmed the numerical stability of the model. Additionally, the model is capable of generating random fractures, which are used directly in flow and dissolution computations. These fractures vary in shape and size and are not constrained by the underlying computational grid.

A high-performance computing environment is essential for achieving long-term stability in water table simulations. The calculations were performed on a platform using a Chinese Hygon C86 7185 32-core processor, running CentOS Linux 7. For these simulations, we utilized 8 cores on a single compute node.

**S 2 Test of the water table in a homogeneous fractured aquifer**

To verify the numerical model, we first compare the results for the homogeneous network with MODFLOW (Harbaugh et al., 2000) and the analytical solution derived using the Dupuit assumption, which can be expressed as follows:

$$H = \sqrt{H_0^2 - \frac{w}{K_x}X^2 + \frac{w\,S}{K_x}X} \tag{S1}$$

where $H_0$ is the river base level, $K_x$ is the equivalent horizontal conductivity in m/d, $S$ is the aquifer length, $X$ is the distance from the left river boundary, and $w$ is the intensity of rainfall recharge.

The homogeneous fracture network is shown in Figure S2.1. We assume translational symmetry and therefore use 2D domain populated by fracture. The horizontal dimension of the domain is 1000 m and vertical 400 m. The distance between the fractures in both the X direction and the Y direction is 10 m. The aperture is 0.01 cm for all the fractures. Along the two side boundaries, nodes lower than 200 m in height are given a constant water head of 200 m. Nodes above 200 m have seepage boundary conditions. The two recharge conditions were tested at 400 mm/a and 800 mm/a.

Within the MODFLOW validation, we calculate the equivalent horizontal $K_x$ and vertical $K_y$ by treating the aquifer as having confined water head boundaries in the X and Y directions, respectively. The conductivities are proportional to the ratio between the resulting flux and the head difference. The horizontal and vertical K values are virtually identical, both approximately 0.00705 m/d. The DRAIN module in MODFLOW was used to model the seepage face by setting each boundary node above the constant head with drainage function, which worked as its water head became higher than the drain's elevation. In Figure S2.1, water table nodes are labeled blue and the corresponding heads have orange labels. The water tables obtained via the Dupuit assumption are always lower than the simulated water table since the vertical flow and seepage face are not considered. Note that the heads at water table nodes are higher than their elevation, but below the elevation of the nearest dry node above them. The water tables simulated with our method are nearly the same as the MODFLOW simulation. Additionally, we can see the seepage face boundaries on both sides, which evidently do not exist in the analytical solution. The analysis demonstrates the effectiveness of our algorithm.

[Figure]

**Figure S2.1:** Water tables and corresponding heads validation with MODFLOW and Dupuit analytical models for two distributed uniform recharge conditions.

**S3 Modeling a water table in a heterogeneous fractured aquifer**

The next step is to test the solution for heterogeneous network. We use the same setting as for the homogenous network but with random generation of fractures and two recharge conditions, 200 mm/a and 400 mm/a. The equivalent horizontal and vertical K values are 0.00469 m/d and 0.00434 m/d, respectively. The number of outer iterations in all evolving time steps varied from 5 to 35. The process of searching for a water table takes approximately 3 to 4 hours during the initial modeling stages of karst evolution, and it is performed on a high-performance computing platform that utilizes 8 cores.

The fracture flow and water table data are shown in Figure S3.1 and Figure S3.2. The water table is discontinuous because of the inhomogeneous distribution of fractures. Only a few nodes for the simulated water table are lower than the analytical water table. The difference between the elevation and head at water table nodes varies due to the heterogeneity of the network. The seepage faces above the constant head boundaries on the both sides of the domain, are successfully simulated as the Signorini boundary (Jiang et al., 2013). Considering these two recharge conditions, the algorithm performed well in modelling the water table in heterogeneous network.

[Figure]

**Figure S3.1.** Modelling the phreatic flow in random fractures under 400mm/a and 200mm/a distributed uniform recharge conditions.

[Figure]

**Figure S3.2:** The simulated and analytical results of random fracture water tables and water heads of corresponding nodes.

**S4 Changes of concentration within individual fracture and on a network scale**

S4.1 Dissolution and change of concentration along an individual fracture

[Figure]

**Figure S4.1:** Calculating $Ca^{2+}$ concentration along one single fracture and at fractures joint node and near the water table.

To calculate change of concentration within a single fracture, we use a Lagrangian approach and imagine a water parcel with volume $dV=Pdx$ moving along the fracture with velocity $v=Q/A$, where A is a flow crosssection, and calculate the change of concentration within the parcel (Figure S4.1). In a time $dt$ the change of concentration in a parcel is equal to:

$$dC = F(c) \cdot P \cdot dx \cdot dt/dV \tag{S2}$$

Where $P$ is flow perimeter and $Pdx$ is the surface area of water/rock contact. Rearranging the equation gives:

$$\int_{C_i}^{c(x)} \frac{dc}{F(c)} = \frac{P}{Q}x = \frac{2(b+w)}{qw}x \approx \frac{2}{q}x \tag{S3}$$

The last term is an approximation for a wide fracture with the lateral width $w$ and aperture $b$, where $w>>b$; $q$ is flow rate per unit width. Using rate Equations (Eq. 7 and Eq. 8) for $F(c)$, we get:

$$C(x) = C_{eq} - \left(C_{eq} - C_i\right)e^{-\left(\frac{2k_1}{q\,C_{eq}}x\right)}, \quad (C < C_s) \tag{S4}$$

$$C(x) = C_{eq} - C_{eq}\left(C_{eq} - C_i\right)\sqrt[3]{\frac{q\,C_{eq}}{\left(C_{eq} - C_i\right)^3 6k_4x + q\,C_{eq}^{\,4}}}, (C_s < C < C_{eq}) \tag{S5}$$

Where $k_1$ and $k_4$ are rate constants. $C_{eq}$ and $C_s$ are the equilibrium concentration and the switch concentration of $Ca^{2+}$ ions.

The change of concentration $\Delta C$ at the outlet of the fracture is given by:

$$\Delta C = C_{eq} - C_{eq}e^{-\left(\frac{2k_1}{q\,C_{eq}}L\right)} + C_i\left(e^{-\left(\frac{2k_1}{q\,C_{eq}}L\right)} - 1\right), (\,C_i<C_s, x_s>L) \tag{S6}$$

$$\Delta C = C_{eq} - C_i - 0.1\,C_{eq}^2\sqrt[3]{\frac{q\,C_{eq}}{(0.1C_{eq})^3 6k_4L + q\,C_{eq}^{\,4}}}, (C_i<C_s, x_s<L) \tag{S7}$$

$$\Delta C = C_{eq} - C_i - C_{eq}^2\sqrt[3]{\frac{q\,C_{eq}}{(C_{eq}-C_i)^3 6k_4L + q\,C_{eq}^{\,4}}} + C_{eq}C_i\sqrt[3]{\frac{q\,C_{eq}}{(C_{eq}-C_i)^3 6k_4L + q\,C_{eq}^{\,4}}}, (\,C_i>C_s) \tag{S8}$$

Where $x_s$ is the switch distance of $C_s$. If $C_i < C_s$ and $x_s > L$, dissolved mass is calculated from Eq. (S6). If $C_i < C_s$ and $x_s < L$, Eq. (S7) is used. If $C_i > C_s$, Eq. (S8) is used for dissolved mass directly.

S4.2 Following concentration at the network scale

To assure that concentrations at the input nodes are always known, we follow the procedure of Siemers and Dreybrodt (1998) and Gabrovšek and Dreybrodt (2000). The process begins at the network's boundary nodes with the highest hydraulic heads, where head or flux values and concentrations are prescribed. Calculations then proceed sequentially along the hydraulic gradient. As illustrated in Figure S4.1(b and c), the concentration $C_j$ at the node $j$ is calculated using the complete mixing assumption. This involves computing the flow-weighted average of the incoming concentrations:

$$c_j = \frac{\sum_i C_{ij}^{out} q_{ij} + \sum_k C_0 q_{wk}}{\sum_i q_{ij} + \sum_k q_{wk}} \tag{S9}$$

Where $q_{wk}$ and $C_0$ are flow and concentration of direct recharge at water table nodes; $q_{ij}$ and $C_{ij}^{out}$ are the output flow and concentration of fractures connecting nodes $i$ and $j$; $i$ sums over confined nodes that deliver flow to j, and k runs over direct input at the water table.

**References**

Harbaugh, A. W., Banta, E. R., Hill, M. C., and McDonald, M. G.: MODFLOW-2000, the US Geological Survey modular ground-water model: User guide to modularization concepts and the ground-water flow process, U.S. GEOLOGICAL SURVEY, Open-File Report 00-92, https://doi.org/10.3133/ofr200092, 2000.

Jiang, Q., Yao, C., Ye, Z., and Zhou, C.: Seepage flow with free surface in fracture networks, Water Resour. Res., 49, 176-186, https://doi.org/10.1029/2012WR011991, 2013.

---

## Author Response (AR2)

**REVISION NOTES**

We sincerely thank the reviewers for recognizing the progress made in the revised manuscript and for their further evaluation of our work. We did our best to address the unresolved issues given in the comments. As suggested by Reviewer 2, we have also revised the text to assure consistent flow between the revised and original sections. All changes are highlighted in red for clarity.

We are grateful to the editor and reviewers for your valuable input, and we hope that the revised version of the manuscript meets your expectations.

Sincerely,

The Authors

**POINT-TO-POINT REPLIES TO COMMENTS OF REVIEWER #1**

The authors have made significant revisions to the paper, and the new version is significantly better; however, I still feel that several comments were not fully addressed.

**General Comment 1:** Although there is a "good agreement" between MODFLOW and the authors' solver, details are needed regarding the grid size sensitivity, convergence, and stability of the numerical code, as these are standard practices. That is why the added S1 does not address my comment. Stating that "We also explored the model's behaviour under different flow regimes by simulating the coexistence of turbulent and laminar flows" is not sufficient; one should show the outcome of this exploration in a quantitative way, how many iterations were required to solve the flow equation? How well did they reproduce the benchmark of MODFLOW?, etc.,

**Reply:** We have taken additional steps to evaluate the performance of our model. Specifically, we varied grid sizes and recharge rates and calculated the root mean square difference (RMS) in water heads at water table nodes between our solution and that obtained with MODFLOW (Harbaugh et al., 2000). The results are summarized in Table S1. Three grid sizes and three recharge rates were tested while keeping the aperture fixed. To ensure comparability between models, we used an equivalent hydraulic conductivity in MODFLOW and the corresponding discretization. The RMS difference of water heads is approximately one quarter of the grid size and increases with recharge rate as shown in Table S1. We have updated the Supplement with text and Table.

$$RMS = \sqrt{\frac{1}{n} \sum_{i=1}^{n} (y_i - \hat{y}_i)^2},$$
 (S2)

where  $y_i$  is the MODFLOW water head, and  $\hat{y}_i$  is our method's water head at a water table node, and n is the number of water heads.

Table S1 The model fit between our method and the MODFLOW solution with different grid sizes and recharge rates

| aperture(cm) | grid size (m) | equivalent hydraulic conductivity K (m/d) | recharge rate
(mm/a) | layer-by-
layer
iterations | RMS difference of
water heads at water
table nodes (m) |
|--------------|---------------|-------------------------------------------|-------------------------|----------------------------------|--------------------------------------------------------------|
| 0.01         | 10            | 0.00705                                   | 200                     | 19                               | 1.27                                                         |
| 0.01         | 10            | 0.00705                                   | 400                     | 23                               | 2.45                                                         |
| 0.01         | 10            | 0.00705                                   | 800                     | 34                               | 2.77                                                         |
| 0.01         | 15            | 0.00473                                   | 400                     | 20                               | 2.11                                                         |
| 0.01         | 20            | 0.00353                                   | 400                     | 19                               | 5.07                                                         |

Figure S2.2: The iterative solution of turbulent and laminar flow (Jiao et. al, 2022).

Although turbulence is not present in the cases discussed in this work, the model accounts for the possible transition to turbulent flow. More details on this can be found in Jiao et al. (2022). Figure S2.2 illustrates a network exhibiting the coexistence of laminar and turbulent flow. For a typical case involving approximately 3,000 fractures, the number of iterations in the Newton–Raphson scheme is about two when the entire domain remains in the laminar regime. At the onset of turbulence, the number of iterations increases sharply to around 15. During the turbulent regime, the number of iterations is kept relatively low (below 10) by maintaining a sufficiently small evolution time step—that is, by ensuring that hydraulic properties change only slightly within each time step.

General Comment 2: In the same sense, stating that this is just one realization out of many other possibilities in response to my comment on the relatively high fracture length does not really address my comment. Every realization is one out of many possibilities, which is what makes it a realization from a statistical standpoint. That is why we aim to simulate a "representative case" and justify why it is representative, which was my request. Otherwise, we deal with a toy model intended to outline a process rather than reproduce it. This is even more important when the fracture lengths clearly dominate the whole regime, in a way that may suggest that the results presented are not representative. This should be further clarified.

**Reply:** Thank you for pointing this out. The fracture lengths presented in Table 1 are indeed misleading, as they show the values *before segmentation by the random procedure*. In reality, the network consists of 7,401 fracture segments, with lengths ranging from 0.01 m to 143 m. The average segment length is 15.74 m. We agree that the configuration of the initial fracture network strongly influences the evolution of the aquifer. To assess this sensitivity, we tested several realizations (Figure S5.1) and obtained conceptually identical results.

Characterizing a fracture network for a specific field site remains a significant challenge. In this study, we employed field-relevant fracture orientations, while the initial aperture and length distributions were assigned heuristically, consistent with approaches used by other researchers.

Figure S5.1: The other two random realizations of karstification in the interfluve aquifer.

**Replies to specific comments**

1. I believe that the Yes /No aspect is missing in the "No nodes wet?" part of the chart.

**Reply:** We have corrected the Figure.

Figure S1.1: Flowchart of the flow solution.

2. "This approximation is valid for situations where the solutions are close to equilibrium, which is mostly the case in the presented scenarios". Can you provide a reference to this approximation?

**Reply:** Chemical analyses of the borehole samples indicate that the water is nearly saturated with respect to calcite (Yuan et al., 2002; Wan et al., 1999), as shown in Figure S5.2. Numerous studies have reported high saturation levels of infiltrating water within the vadose zone (e.g., Fairchild & Baker, 2012). Furthermore, during the initial stages of karst aquifer development, equilibrium with calcite is typically achieved over very short distances (Dreybrodt et al., 2005). Thus, even when the initial calcium concentration is low, saturation conditions—under which such an approximation remains valid—are usually reached within a few meters.

Figure S5.2: The  $CO_2$  partial pressure in the soil at the Luojiaao observation site and the  $Ca^{2+}$  content in 3 boreholes and 2 springs in the Luojiaao interfluve aquifer (Yuan et al., 2002; Wan et al., 1999)

3. Figure 7. It seems that the heat map range does not cover the entire range (the red color is missing from the heat map).

**Reply:** We appreciate the observation. We have redrawn the map.

Figure 6. Evolution of fracture apertures under natural karstification at 0 ka (thousand years) (a), at 10 ka (b) and at 20 ka (c).

**References:**

Dreybrodt, W., Gabrovšek, F., and Romanov, D.: Processes of speleogenessis: a modelling approach, Založba ZRC, 2005.

Fairchild, I. J., and Baker, A.: Speleothem science: from process to past environments, John Wiley & Sons, 2012.

Harbaugh, A. W., Banta, E. R., Hill, M. C., and McDonald, M. G.: MODFLOW-2000, the US Geological Survey modular ground-water model: User guide to modularization concepts and the ground-water flow process, U.S. GEOLOGICAL SURVEY, Open-File Report 00-92, https://doi.org/10.3133/ofr200092, 2000.

Jiao, Y., Huang, Q., and Yu, Q.: Influence of initial fractures on the occurrence of karst turbulent flow, Carsologica Sinica, 41, 501-510, (in Chinese), https://doi.org/10.11932/karst20220401, 2022.

Wan, J., Chao, N., Shen, J., and Cai, J.: Study on the carbon cycle in a karst system at the Luojiaao interfluve along the Qingjiang River of western Hubei, China, Carsologica Sinica, 18, 123-128, (in Chinese), https://doi.org/10.3969/j.issn.1001-4810.1999.02.004, 1999.

Yuan, D., Liu, Z., Lin, Y., Shen, J., He, S., Xu, S., Yang, L., Li, B., Qin, J., Cai, W., Cao, J., Zhang, M., Jiang, Z., and Zhao, J.: Karst dynamic system of China, Geological Publishing House, Beijing, (in Chinese), 2002.

**POINT-TO-POINT REPLIES TO COMMENTS OF REVIEWER #2**

The authors have sufficiently addressed my comments and I think the manuscript improved significantly. I only have a few minor comments regarding some of the modifications.

1. Please check carefully the language and style of the revised new sections, as they are often read very differently compared to the original parts, and break the flow of the text.

**Reply:** We appreciate this observation. We have carefully reviewed the language throughout the manuscript and made additional edits to improve readability and ensure a smooth, consistent flow between the revised and original sections. In addition to numerous minor corrections (all highlighted in red in the revised manuscript), we have made some moderate changes as listed below:

- We moved the paragraph at Line 170 (First revision) before the description of algorithm (now Line 150),
- We reformulated Section 3.1 to make it more concise and readable, and to avoid repetitions,
- We moved the paragraph between lines 275 and 285 (First revision, beginning with
   "Karstification represents a form...") to the Discussion and conclusion section (now Lines 350 to 360, as it discusses the results in a broader context.
- We deleted the paragraph (lines 350-355 in first revision), and included its message into
   Discussion and conclusion section.
- To avoid unnecessary segmentation, we have merged the Discussion and Conclusion sections into a single unified section. These previously separate sections included overlapping content that logically belonged to both.
- We have also removed a few redundant sentences that repeated similar content (see also reply to Comment 8).

Importantly, we ensured that all changes were limited to improving style and readability. No reviewed content has been altered.

2. Figure 1. - caption is a bit unclear: what do you mean here by flow solution? In b) and c) you talk about processes but here about modeling? Please clarify this.

**Reply:** We agree that the original wording may have been somewhat misleading. Figure 1a illustrates the position of the water table and the flow lines, which indicate the presence of a stagnant flow zone at the groundwater divide. This is a fundamental outcome of the flow field solution in such settings. To avoid confusion, we have revised the figure caption and removed the reference to "flow solution."

3. Figure 2 - The flowchart really helps with the understanding with the methodology. It was a great idea to link the flowchart elements to the different sections, but then do it for section 2.1 as well.

**Reply:** Thank you for pointing this out. We have corrected the figure accordingly.

4. The text in the figure could be a bit reduced in size. I would also recommend a more figure friendly font (e.g. Arial - see the new flowchart in the supplements (Fig. R2) which reads a bit nicer due to this).

**Reply:** We agree. Done as suggested.

5. Setion 2.2.1 - Don't forget the reference to the new supplementary figure (R2) and the inclusion of the figure to the supplements.

**Reply:** Thank you for a reminder. We have included the new revised figures (Figure S1.1, Table S1, Figure S2.2, Figure S5.1 and Figure S5.2) in the supplements.

6. Section 2.4 I would reduce this part significantly, because it currently reads a bit strange. Also, because this only refers to the flow solution it could be just added to the end of section 2.2.2 - because here it is confusing as a verification to the karstification too. A shorter version could be:

"The numerical model was verified successfully against a MODFLOW model and an analytical model using the Dupuit assumption. Please see the Supplements for details."

**Reply:** We agree and have implemented the reviewer's suggestion. The proposed text has been incorporated *verbatim* at the end of Section 2.2.2, (Line 165).

7. L394: Do you mean that different possible initial aquifer structures can be tested out using the numerical model, to see their impacts later? If so you may use the approach for scenario analysis or even assessing the uncertainties in karst evolution.

**Reply:** Yes, different realizations can indeed be used to explore the parameter space. We have tested several cases and added the results of two additional realizations to the Supplementary Material (Figure S5.1; see also our reply to Reviewer 1). We agree that further work in this direction would be valuable to broaden the concept, and we plan to pursue this in future studies.

8. L402: "These low karstified..." - this sentence really stands out from here, I think you can delete it (too introductory to the end of the paper and also breaks the linkage between the sentences before and after).

**Reply:** We agree with the comment and have deleted the sentence.